# Slow oscillation–spindle coupling strength predicts real-life gross-motor learning in adolescents and adults

Michael A Hahn[1,2,3]*, Kathrin Bothe[1,2], Dominik Heib[1,2], Manuel Schabus[1,2], Randolph F Helfrich[3], Kerstin Hoedlmoser[1,2]*

[1]Department of Psychology, Laboratory for Sleep, Cognition and Consciousness Research, University of Salzburg, Salzburg, Austria; [2]Centre for Cognitive Neuroscience Salzburg (CCNS), University of Salzburg, Salzburg, Austria; [3]Hertie-Institute for Clinical Brain Research, University Medical Center Tübingen, Tübingen, Germany

**Abstract** Previously, we demonstrated that precise temporal coordination between slow oscillations (SOs) and sleep spindles indexes declarative memory network development (Hahn et al., 2020). However, it is unclear whether these findings in the declarative memory domain also apply in the motor memory domain. Here, we compared adolescents and adults learning juggling, a real-life gross-motor task. Juggling performance was impacted by sleep and time of day effects. Critically, we found that improved task proficiency after sleep lead to an attenuation of the learning curve, suggesting a dynamic juggling learning process. We employed individualized cross-frequency coupling analyses to reduce inter- and intragroup variability of oscillatory features. Advancing our previous findings, we identified a more precise SO–spindle coupling in adults compared to adolescents. Importantly, coupling precision over motor areas predicted overnight changes in task proficiency and learning curve, indicating that SO–spindle coupling relates to the dynamic motor learning process. Our results provide first evidence that regionally specific, precisely coupled sleep oscillations support gross-motor learning.

**\*For correspondence:**
michael.andreas.hahn@gmail.com (MAH);
kerstin.hoedlmoser@plus.ac.at (KH)

**Competing interest:** The authors declare that no competing interests exist.

## Editor's evaluation

The authors used a clever design, in which adolescents and adults learned to juggle, to study the impact of sleep and associated oscillations on the consolidation of motor memory across age groups. Overall, the topic and the results of the present study are interesting and timely, and extends previous findings in the declarative memory domain to the motor memory domain.

## Introduction

Sleep actively supports learning (**Diekelmann and Born, 2010**). The influential active system consolidation theory suggests that long-term consolidation of memories during sleep is driven by a precise temporal interplay between sleep spindles and slow oscillations (SOs; **Diekelmann and Born, 2010**; **Klinzing et al., 2019**). Memories acquired during wakefulness are reactivated in the hippocampus during sharp-wave ripple events in sleep (**Wilson and McNaughton, 1994**; **Zhang et al., 2018**). These events are nested within thalamocortical sleep spindles that mediate synaptic plasticity (**Niethard et al., 2018**; **Rosanova and Ulrich, 2005**). Sleep spindles in turn are thought to be facilitated by the depolarizing phase of cortical SOs thereby forming SO–spindle complexes during which the subcortical–cortical network communication is optimal for information

transfer (*Chauvette et al., 2012*; *Clemens et al., 2011*; *Helfrich et al., 2019*; *Helfrich et al., 2018*; *Latchoumane et al., 2017*; *Mölle et al., 2011*; *Ngo et al., 2020*; *Niethard et al., 2018*; *Schreiner et al., 2021*; *Staresina et al., 2015*).

Several lines of research recently demonstrated that precisely timed SO–spindle interaction mediates successful memory consolidation across the lifespan (*Hahn et al., 2020*; *Helfrich et al., 2018*; *Mikutta et al., 2019*; *Mölle et al., 2011*; *Muehlroth et al., 2019*). Critically, SO–spindle coupling as well as spindles and SOs in isolation are related to neural integrity of memory structures such as medial prefrontal cortex, thalamus, hippocampus, and entorhinal cortex (*Helfrich et al., 2021*; *Helfrich et al., 2018*; *Ladenbauer et al., 2017*; *Mander et al., 2017*; *Muehlroth et al., 2019*; *Spanò et al., 2020*; *Winer et al., 2019*). Thus, converging evidence suggests that SO–spindle coupling does not only actively transfer mnemonic information during sleep but also indexes general efficiency of memory pathways (*Helfrich et al., 2021*; *Mander et al., 2017*). In our recent longitudinal work, we found that SO–spindle coordination was not only becoming more consistent from childhood to late adolescence but also directly predicted enhancements in declarative memory formation across those formative years (*Hahn et al., 2020*). However, because the active system consolidation theory assumes a crucial role of hippocampal memory replay for sleep-dependent memory consolidation, most studies, including our own, focused on the effect of SO–spindle coupling on hippocampus-dependent declarative memory consolidation. Therefore, the role of SO–spindle coordination for motor learning or consolidation of procedural information remains poorly understood.

While sleep's beneficial role for motor memory formation has been extensively investigated and frequently related to individual oscillatory activity of sleep spindles and SO (*Barakat et al., 2011*; *Boutin et al., 2018*; *Fogel et al., 2017*; *Huber et al., 2004*; *King et al., 2017*; *Nishida and Walker, 2007*; *Pinsard et al., 2019*; *Tamaki et al., 2013*; *Tamaki et al., 2008*; *Vahdat et al., 2017*; *Walker et al., 2002*), there is little empirical evidence for the involvement of the timed interplay between spindles and SO. In rodents, the neuronal firing pattern in the motor cortex was more coherent during spindles with close temporal proximity to SOs after engaging in a grasping motor task (*Silversmith et al., 2020*). In humans, stronger SO–spindle coupling related to higher accuracy during mirror tracing, a motor adaption task where subjects trace the line of a shape while looking through a mirror (*Mikutta et al., 2019*). So far, research focused on laboratory suitable fine-motor sequence learning or motor adaption tasks, which has hampered our understanding of memory consolidation for more ecologically valid gross-motor abilities that are crucial for our everyday life (for a review see *King et al., 2017*).

Only few studies have investigated the effect of sleep on complex real-life motor tasks. Overnight performance benefits for riding an inverse steering bike have been shown to be related to spindle activity in adolescents and adults (*Bothe et al., 2019*; *Bothe et al., 2020*). Similarly, juggling performance was supported by sleep and juggling training induced power increments in the spindle and SO frequency range during a nap (*Morita et al., 2012*; *Morita et al., 2016*). Remarkably, juggling has been found to induce lasting structural changes in the hippocampus and midtemporal areas outside of the motor network (*Boyke et al., 2008*; *Draganski et al., 2004*), making it a promising expedient to probe the active system consolidation framework for gross-motor memory. Importantly, this complex gross-motor skill demands accurately executed movements that are coordinated by integrating visual, sensory, and motor information. Yet, it remains unclear whether learning of these precisely coordinated movements demand an equally precise temporal interplay within memory networks during sleep.

Previously, we demonstrated that SO and spindles become more tightly coupled across brain maturation which predicts declarative memory formation enhancements (*Hahn et al., 2020*). Here, we expand on our initial findings by investigating early adolescents and young adults learning how to juggle as real-life complex gross-motor task. We first sought to complete the picture of SO–spindle coupling strength development across brain maturation by comparing age ranges that were not present in our initial longitudinal dataset. Second, we explicitly tested the assumption that precisely coordinated SO–spindle interaction supports learning of coordinated gross-motor skills.

By leveraging an individualized cross-frequency coupling approach, we demonstrate that adults have a more precise interplay of SO and spindles than early adolescents. Importantly, the consistency of the SO–spindle coupling dynamic tracked the dynamic learning process of a gross-motor task.

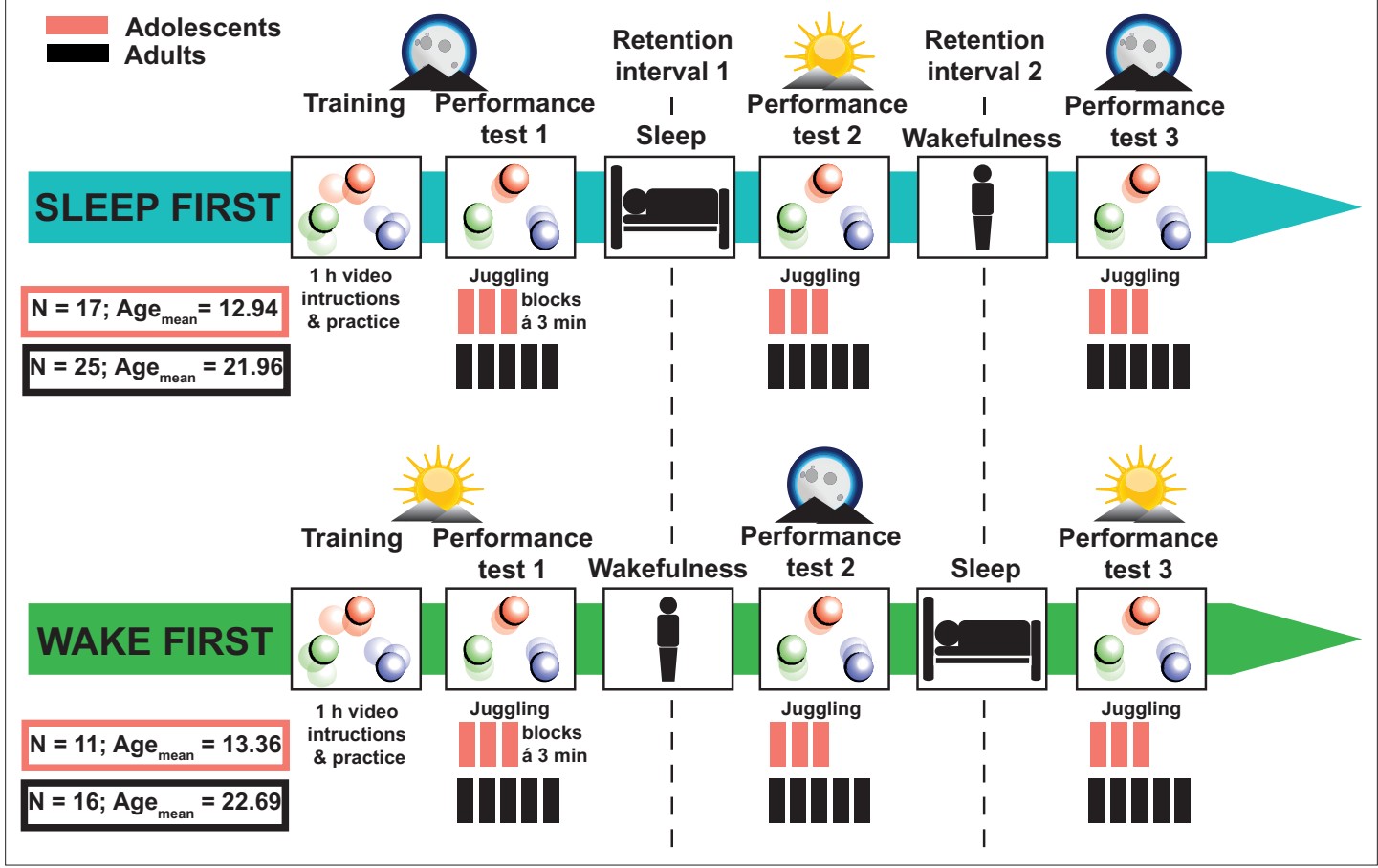

**Figure 1.** Study design. Adolescents (*N* = 28; 23 males) and adults (*N* = 41; 25 males) without prior juggling experience were divided into *sleep-first* and *wake-first* groups. Participants in the *sleep-first* group trained to juggle for 1 hr with video instructions in the evening. Juggling performance was tested before and after a retention interval containing sleep (1), followed by a third juggling test after a retention interval containing wakefulness (2). Participants in the *wake-first* group followed the same protocol but in reverse order (i.e., training in the morning, first retention interval containing wakefulness and second retention interval containing sleep). Polysomnography was recorded during an adaptation night and a learning night at the respective sleep retention interval. Psychomotor vigilance tasks were conducted before each performance test. Adolescents only performed three juggling blocks per test to avoid a too excessive training load.

## Results

Healthy adolescents (*n* = 28, age: 13.11 ± 0.79 years, mean ± standard deviation [SD]) and young adults (*n* = 41, age: 22.24 ± 2.15 years) performed a complex gross-motor learning task (juggling) before and after a full night retention interval as well as before and after a retention interval during wakefulness (*Figure 1*). To assess the impact of sleep on juggling performance, we divided the participants into a *sleep-first* group (i.e., sleep retention interval followed by a wake retention interval) and a *wake-first* group (i.e., wake retention interval followed by a sleep retention interval). Polysomnography (PSG) was recorded during an adaptation night and during the respective sleep retention interval (i.e., learning night) except for the adult *wake-first* group (for sleep architecture descriptive parameters of the adaptation night and learning night as well as for adolescents and adults see *Supplementary file 1*—tables 1 and 2). Participants without prior juggling experience trained to juggle for 1 hr. We measured the amount of successful three-ball cascades (i.e., three consecutive catches) during performance tests in multiple 3-min blocks (3 × 3 min for adolescents; 5 × 3 min for adults) before and after the respective retention intervals. Adolescents performed fewer blocks than adults to alleviate exhaustion from the extensive juggling training.

## Behavioral results: juggling performance and disentangling the learning process

Adolescents improved their juggling performance over the course of all nine blocks (*Figure 2A*, top; $F_{3.957, 94.962} = 6.948$, p < 0.001, $\eta^2 = 0.23$). There was neither an overall difference in performance between the *sleep-first* and *wake-first* groups ($F_{1, 24} = 1.002$, p = 0.327, $\eta^2 = 0.04$), nor did they differ over the course of the juggling blocks ($F_{3.957, 94.962} = 1.148$, p = 0.339, $\eta^2 = 0.05$). Similar to the adolescents, adults improved in performance across all 15 blocks (*Figure 2B*, top; $F_{4.673, 182.241} = 11.967$, p < 0.001, $\eta^2 = 0.24$), regardless of group ($F_{4.673, 182.241} = 0.529$, p = 0.742, $\eta^2 = 0.01$). Further, there was no overall difference in performance between the *sleep-first* and *wake-first* groups in adults ($F_{1, 39} = 1.398$, p = 0.244, $\eta^2 = 0.04$). Collectively, these results show, that participants do not reach asymptotic level juggling performance (for single subject data of good and bad performers, see *Figure 2—figure supplement 1A, B*). In other words, the gross-motor skill learning process is still in progress in adolescents and adults. Therefore, we wanted to capture the progression of the learning process, rather than absolute performance metrics (i.e., mean performance) that would underestimate the dynamics of gross-motor learning.

Since subjects did not reach asymptotic level performance, but learning was ongoing, we parameterized the juggling learning process by estimating the learning curve for each performance test using a first-degree polynomial fit to the different blocks (*Figure 2A–C*, black lines). We considered the slope of the resulting trend as learning curve. The learning process of complex motor skills is thought to consist of a fast initial learning stage during skill acquisition and a much slower skill retaining learning stage (*Dayan and Cohen, 2011*; *Doyon and Benali, 2005*). In other words, within-learning session performance gains are rapid at the beginning, but taper off with increased motor skill proficiency, resembling a power-law curve. Therefore, we also estimated the task proficiency per performance test at the first time point as predicted by the model, since the learning curve is expected to be influenced by the individual juggling aptitude. Importantly, the estimated task proficiency was comparable to the observed values in the corresponding first juggling block (performance test 1: $rho_s = 0.98$, p < 0.001; performance test 2: $rho_s = 0.97$, p < 0.001). Besides having a more accurate picture of juggling performance, this parameterization also allowed us to compare performance of adolescents and adults on a similar scale because of the different number of juggling blocks. A mixed ANOVA with the factors performance test (pre- and postretention interval), condition group (*sleep-first* and *wake-first*) and age group (adolescents and adults) showed a significant interaction between performance test and condition group ($F_{1, 65} = 4.868$, p = 0.031, $\eta^2 = 0.07$). This result indicates that regardless of age, the juggling learning curve becomes steeper after sleep than after wakefulness, thus indicating that sleep impacts motor learning (*Figure 2D*). No other interactions or main effects were significant (for the complete ANOVA report, see *Supplementary file 1—table 3*). When analyzing the task proficiency before and after the first retention interval, depending on condition and age group, we found a significant interaction between condition and age group (*Figure 2E*; $F_{1, 65} = 5.210$, p = 0.026, $\eta^2 = 0.07$), showing that the adult *sleep-first* group had better overall task proficiency than the *wake-first* group, whereas the adolescent *sleep-first* group was worse than the *wake-first* group. The interaction (performance test × condition group) did not reach significance ($F_{1, 65} = 1.882$, p = 0.175, $\eta^2 = 0.03$; also see *Supplementary file 1—table 4*). Collectively, these results suggest that sleep influences learning of juggling as a gross-motor task.

*Figure 2A, B* indicates that performance tests in the morning might be characterized by a steeper learning curve than the evening tests. We confirmed this observation using a linear mixed model (*Supplementary file 1—table 5A, B*). While this finding might also indicate a circadian influence on learning in our task, we did not find evidence for a circadian effect on sensitive psychomotor vigilance task reaction times. Neither when comparing sleep-first and wake-first groups (*Figure 2—figure supplement 1C*), nor when specifically probing evening and morning performance tests (*Supplementary file 1—table 5E, F*). However, these analyses cannot exclude all circadian effects. Therefore, we modeled learning curve and task proficiency with time of day (morning session, evening session) and sleep after learning as fixed effects and subjects as random effects to further disentangle circadian and sleep specific effects. Results for learning curve were inconclusive for both fixed effects (time of day: Beta = −1.008, $t(202) = −1.625$, p = 0.106, $CI_{95} = [−2.231, 0.215]$; sleep after learning: Beta = 0.172, $t(202) = 0.268$, p = 0.789, $CI_{95} = [−1.093, 1.437]$; *Supplementary file 1—table 6A*). Task proficiency was overall better in the evening performance tests (Beta = 5.751, $t(202) = 2.252$, p = 0.011, $CI_{95} = $

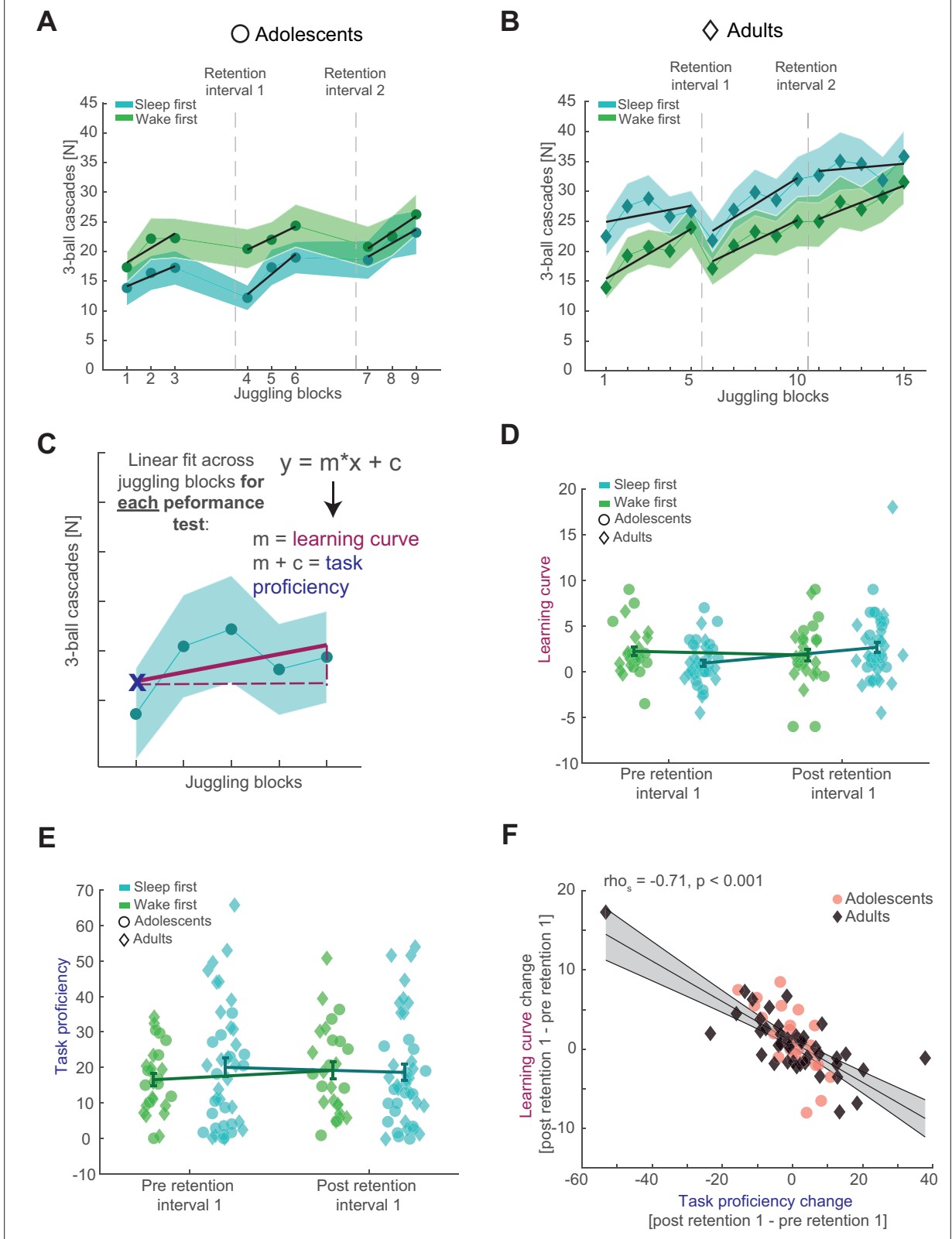

**Figure 2.** Behavioral results and parameterizing juggling performance. (**A**) The number of successful three-ball cascades (mean ± standard error of the mean [SEM]) of adolescents (circles) for the *sleep-first* (blue) and *wake-first* groups (green) per juggling block. Grand average learning curve (black lines) as computed in (**C**) are superimposed. Dashed lines indicate the timing of the respective retention intervals that separate the three performance tests. Note that adolescents improve their juggling performance across the blocks. (**B**) Same conventions as in (**A**) but for adults (diamonds). Similar to

*Figure 2 continued on next page*

*Figure 2 continued*

adolescents, adults improve their juggling performance across the blocks regardless of group. (**C**) Schematic representation of the juggling learning process parameterization. We used a linear fit across all juggling blocks within a performance test to estimate the learning curve (*m*) and the task proficiency (linear line equation solved for *x* = 1) for each corresponding performance test. (**D**) Comparison of the juggling learning curve (mean ± standard error of the mean [SEM]) between the *sleep-first* (blue) and *wake-first* groups (green) of adolescents (circles) and adults (diamonds) before and after the first retention interval to investigate the influence of sleep. Single subject data are plotted in the corresponding group color and age icon. Participants in the *sleep-first* group showed a steeper learning curve than the *wake-first* group after the first retention interval. (**E**) Same conventions as in (**D**) but for the task proficiency metric. Adolescents in the *wake-first* group had better overall task proficiency than adolescents in the *sleep-first* group. Adults in the *sleep-first* group displayed better overall task proficiency than adults in the *wake-first* group. (**F**) Spearman rank correlation between the overnight change in task proficiency (post–preretention interval) and the overnight change in learning curve with robust linear trend line collapsed over the whole sample. Gray-shaded area indicates 95% confidence intervals of the trend line. Adolescents are denoted as red circles and adults as black diamonds. A strong inverse relationship indicated that participants with an improved task proficiency show flatter learning curves.

The online version of this article includes the following figure supplement(s) for figure 2:

**Figure supplement 1.** Additional behavioral results and control analyses.

[1.310, 10.192]) and additionally trended to benefit from sleep after learning (Beta = 3.795, *t*(202) = 1.672, p = 0.096, CI$_{95}$ = [−0.680, 8.271]; *Supplementary file 1*—table 6B). These results suggest that both time of day and sleep contribute to the overall juggling performance.

Next, we further dissected the relationship between changes in the learning curve and task proficiency after the first retention interval. We hypothesized, that a stronger increase in task proficiency across sleep would lead to a flatter learning curve based on the assumption that motor skill learning involves fast and slow learning stages. Indeed, we confirmed a strong negative correlation between the change (postretention values − preretention values) in task proficiency and the change in learning curve after the retention interval (*Figure 2F*; rho$_s$ = −0.71, p < 0.001), which also remained strong after outlier removal (*Figure 2—figure supplement 1D*). This result indicates that participants who consolidate their juggling performance after a retention interval show slower gains in performance. Note, that the flattening of the learning curve does not necessarily indicate worse learning but rather mark a more progressed learning stage. These results demonstrate a highly dynamic gross-motor skill learning process. Given that sleep influences the juggling learning curve, we aimed to determine whether sleep oscillation dynamics track the dynamics of gross-motor learning.

## Electrophysiological results: interindividual variability and SO–spindle coupling

To determine the nature of the timed coordination between the two cardinal sleep oscillations, we adopted the same principled individualized approach we developed earlier (*Hahn et al., 2020*). First, we compared oscillatory power between adolescents and adults in the frequency range between 0.1 and 20 Hz during NREM (2 and 3) sleep, using cluster-based permutation tests (*Maris and Oostenveld, 2007*). Spectral power was elevated in adolescents as compared to adults across the whole tested frequency range (*Figure 3—figure supplement 1A* left for representative electrode Cz; cluster test: p < 0.001, *d* = 1.88). Similar to the previously reported developmental patterns of sleep oscillations from childhood to adolescence (*Hahn et al., 2020*), this difference was explained by a spindle frequency peak shift and broadband decrease in the fractal or 1/f trend of the signal, thus directly replicating and extending our previous findings in a separate sample. After estimating the fractal component of the power spectrum by means of irregular-resampling autospectral analysis (*Wen and Liu, 2016*), we found that adolescents exhibited a higher offset of fractal component on the *y*-axis than adults (*Figure 3—figure supplement 1A* middle; cluster test: p < 0.001, *d* = 1.99). Next, we subtracted the fractal component from the power spectrum, which revealed clear distinct oscillatory peaks in the SO (<2 Hz) and sleep spindle range (11–16 Hz) for both adolescents and adults (*Figure 3—figure supplement 1A*, right). Importantly, we observed the expected spatial amplitude topography with stronger frontal SO and pronounced centroparietal spindles for both age groups (*Figure 3A* left).

Critically, the displayed group averages of the oscillatory residuals (*Figure 3—figure supplement 1A*, right) underestimate the interindividual variability of the spindle frequency peak (*Figure 3A*, right; oscillatory residuals for all subjects at Cz). Even though we found the expected systematic spindle frequency increase in a frontoparietal cluster from adolescence to adulthood (*Figure 3—figure*

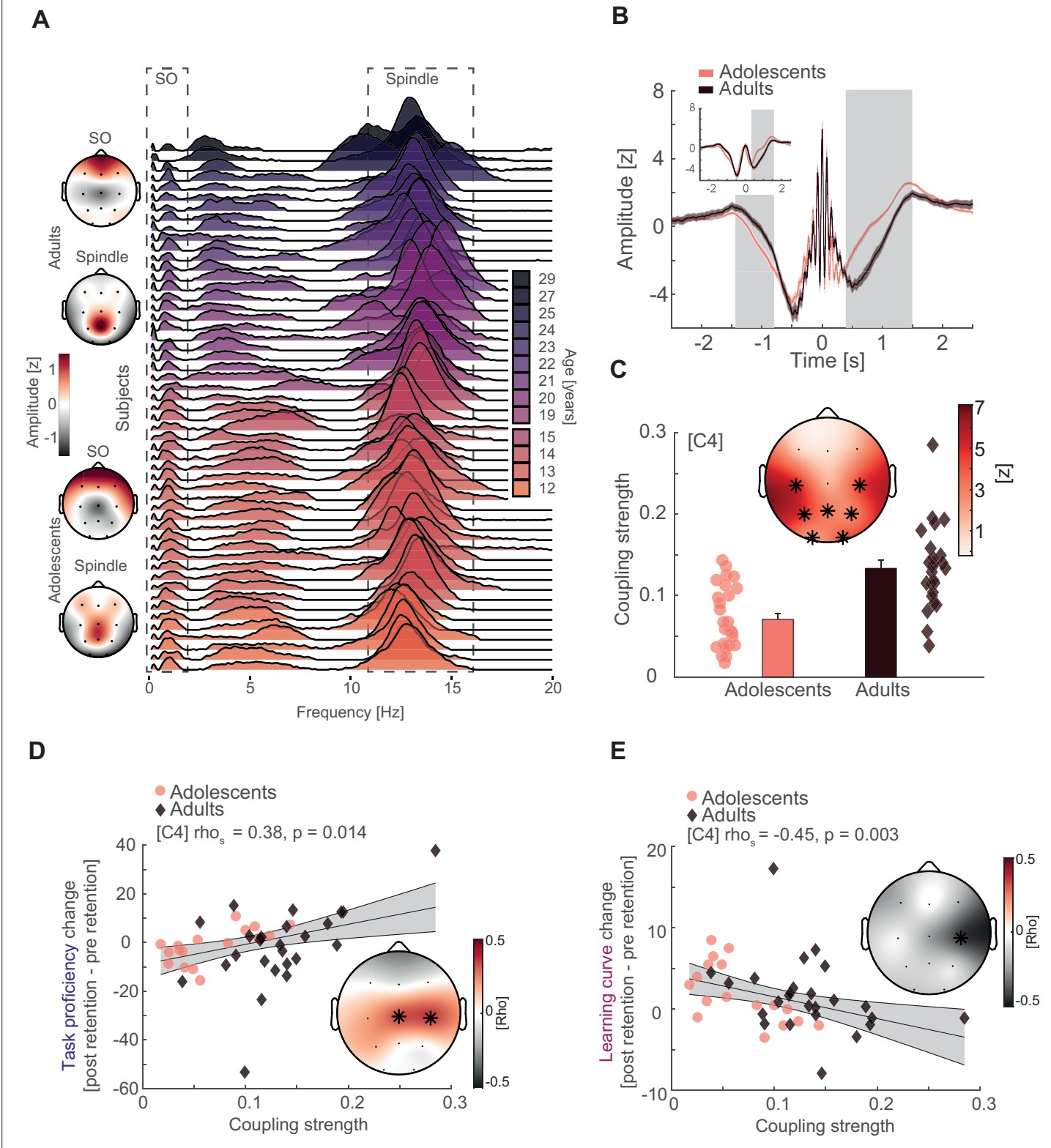

**Figure 3.** Interindividual variability, slow oscillation (SO)–spindle coupling development, and neural correlates of gross-motor learning dynamics. (**A**) Left: topographical distribution of the 1/f corrected SO and spindle amplitude as extracted from the oscillatory residual (*Figure 3—figure supplement 1A*, right). Note that adolescents and adults both display the expected topographical distribution of more pronounced frontal SO and centroparietal spindles. Right: single subject data of the oscillatory residual for all subjects with sleep data color coded by age (darker colors indicate older subjects). SO and spindle frequency ranges are indicated by the dashed boxes. Importantly, subjects displayed high interindividual variability in the sleep spindle

*Figure 3 continued on next page*

*Figure 3 continued*

range and a gradual spindle frequency increase by age that is critically underestimated by the group average of the oscillatory residuals (*Figure 3— figure supplement 1A*, right). (**B**) Spindle peak locked epoch (NREM3, co-occurrence corrected) grand averages (mean ± standard error of the mean [SEM]) for adolescents (red) and adults (black). Inset depicts the corresponding SO-filtered (2 Hz lowpass) signal. Gray-shaded areas indicate significant clusters. Note, we found no difference in amplitude after normalization. Significant differences are due to more precise SO–spindle coupling in adults. (**C**) Top: comparison of SO–spindle coupling strength between adolescents and adults. Adults displayed more precise coupling than adolescents in a centroparietal cluster. *T*-Scores are transformed to *z*-scores. Asterisks denote cluster-corrected two-sided p < 0.05. Bottom: Exemplary depiction of coupling strength (mean ± SEM) for adolescents (red) and adults (black) with single subject data points. Exemplary single electrode data (bottom) is shown for C4 instead of Cz to visualize the difference. (**D**) Cluster-corrected correlations between individual coupling strength and overnight task proficiency change (post–pretraining) for adolescents (red, circle) and adults (black, diamond) of the sleep-first group (left, data at C4). Asterisks indicate cluster-corrected two-sided p < 0.05. Gray-shaded area indicates 95% confidence intervals of the trend line. Participants with a more precise SO–spindle coordination show improved task proficiency after sleep. Note that the change in task proficiency was inversely related to the change in learning curve (*Figure 2F*), indicating that a stronger improvement in task proficiency related to a flattening of the learning curve. Further note that the significant cluster formed over electrodes close to motor areas. (**E**) Cluster-corrected correlations between individual coupling strength and overnight learning curve change. Same conventions as in (**D**). Participants with more precise SO–spindle coupling over C4 showed attenuated learning curves after sleep.

The online version of this article includes the following figure supplement(s) for figure 3:

**Figure supplement 1.** Sleep oscillation features and additional SO-spindle coupling analyses.

**Figure supplement 2.** Supplemental behavioral analyses of the adolescent group, additional coupling strength with behavior correlations, and control analyses.

**Figure supplement 3.** Partial correlations controlling for age, PVT reaction time, and sleep architecture.

**Figure supplement 4.** Partial correlations controlling for sleep oscillation event features.

supplement 1B; cluster test: p = 0.002, d = −0.87), both respective age groups showed a high degree of variability of the interindividual spindle peak.

Based on these findings, we separated the oscillatory activity from the fractal activity for every subject at every electrode position to capture the individual features of SO and sleep spindle oscillations. We then used the extracted individual features from the oscillatory residuals to adjust SO and spindle detection algorithms (*Hahn et al., 2020*; *Helfrich et al., 2018*; *Mölle et al., 2011*; *Staresina et al., 2015*) to account for the spindle frequency peak shift and high interindividual variability. To ensure the simultaneous presence of the two interacting sleep oscillations in the signal, we followed a conservative approach and restricted our analyses to NREM3 sleep given the low co-occurrence rate in NREM2 sleep (*Figure 3—figure supplement 1C, D*) which can cause spurious coupling estimates (*Hahn et al., 2020*). Further, we only considered spindle events that displayed a concomitant detected SO within a 2.5-s time window.

We identified an underlying SO component (2 Hz low-pass filtered trace) in the spindle peak locked averages for adolescents and adults on single subject and group average basis (*Figure 3—figure supplement 1E*), indicating a temporally precise interaction between sleep spindles and SO that is clearly discernible in the time domain.

To further assess the interaction between SO and sleep spindles, we computed SO-trough-locked time–frequency representations (*Figure 3—figure supplement 1F*). Adolescents and adults revealed a shifting temporal pattern in spindle activity (11–16 Hz) depending on the SO phase. In more detail, spindle activity decreased during the negative peak of the SO ('down-state') but increased during the positive peak ('up-state'). This temporal pattern and the underlying SO component in spindle event detection (*Figure 3—figure supplement 1E*) confirm the coordinated nature of the two major sleep oscillations in adolescents and adults.

Next, we determined the coordinated interplay between SO and spindles in more detail by analyzing individualized event-locked cross-frequency interactions (*Dvorak and Fenton, 2014*; *Hahn et al., 2020*; *Helfrich et al., 2019*). In brief, we extracted the instantaneous phase angle of the SO component (<2 Hz) corresponding to the positive spindle amplitude peak for all trials at every electrode per subject. We assessed the cross-frequency coupling based on z-normalized spindle epochs (*Figure 3B*) to alleviate potential power differences due to age (*Figure 3—figure supplement 1A*) or different EEG-amplifier systems that could potentially confound our analyses (*Aru et al., 2015*). Importantly, we found no amplitude differences around the spindle peak (point of SO-phase readout)

between adolescents and adults using cluster-based random permutation testing (*Figure 3B*), indicating an unbiased analytical signal. This was also the case for the SO-filtered (<2 Hz) signal (*Figure 3B*, inset). Critically, the significant differences in amplitude from −1.4 to −0.8 s (p = 0.023, *d* = −0.73) and 0.4–1.5 s (p < 0.001, d = 1.1) are not caused by age-related differences in power or different EEG systems but instead by the increased coupling strength (i.e., higher coupling precision of spindles to SOs) in adults giving rise to a more pronounced SO-wave shape when averaging across spindle peak locked epochs. Further, we specifically focused our analyses on spindle events to account for the higher variability in the spindle frequency band than in the SO band (*Figure 3A*). Based on these adjusted phase values, we derived the coupling strength defined as 1 − circular variance. This metric describes the consistency of the SO–spindle coupling (i.e., higher coupling strength indicates more precise coupling) and has previously been shown to accurately track brain development and memory formation (*Hahn et al., 2020*). As expected, adults had a higher coupling strength in a centroparietal cluster than adolescents (*Figure 3C*; cluster test: p < 0.001, d = 0.88), indicating a more precise interplay between SO and spindles during adulthood.

## SO–spindle coupling tracks gross-motor learning

After demonstrating that SO–spindle coupling becomes more precise from early adolescence to adulthood, we tested the hypothesis, that the dynamic interaction between the two sleep oscillations explains the dynamic process of complex gross-motor learning. When taking the behavioral analyses into account, we did not find any evidence for a difference between the two age groups on the impact of sleep on the learning curve (*Figure 2D*). Therefore, we did not differentiate between adolescents and adults in our correlational analyses. Furthermore, given that we only recorded PSG for the adults in the sleep-first group and that adolescents in the wake-first group showed enhanced task proficiency at the time point of the sleep retention interval due to additional training (*Figure 3—figure supplement 2A*), we only considered adolescents and adults of the *sleep-first* group to ensure a similar level of juggling experience (for summary statistics of sleep architecture and SO and spindle events of subjects that entered the correlational analyses; see *Supplementary file 1*—table 7). Notably, we found no differences in electrophysiological parameters (i.e., coupling strength, event detection) between the adolescents of the wake-first and sleep-first groups (*Figure 3—figure supplement 2B* and *Supplementary file 1*—table 8). To investigate whether coupling strength in the night of the first retention interval explains overnight changes of task proficiency (postretention interval 1 − preretention interval 1), we computed cluster-corrected correlation analyses. We identified a significant central cluster (*Figure 3D*; mean rho = 0.37, p = 0.017), indicating that participants with a more consistent SO–spindle interplay have stronger overnight improvements in task proficiency.

Given that we observed a strong negative correlation between task proficiency at a given time point and the steepness of the subsequent learning curve (*Figure 2F*) as subjects improve but do not reach ceiling level performance, we conversely expected a negative correlation between learning curve and coupling. Given this dependency, we observed a significant cluster-corrected correlation at C4 (*Figure 3E*; $rho_s$ = −0.45, p = 0.039, cluster-corrected), showing that participants with a more precise SO–spindle coupling exhibit a flatter learning curve overnight. This observation is in line with a trade-off between proficiency and learning curve, which exhibits an upper boundary (100% task proficiency). In other words, individuals with high performance exhibit a smaller gain through additional training when approaching full task proficiency.

Critically, when computing the correlational analyses separately for adolescents and adults, we identified highly similar effects at electrode C4 for task proficiency (*Figure 3—figure supplement 2C*) and learning curve (*Figure 3—figure supplement 2D*) in each group. These complementary results demonstrate that coupling strength predicts gross-motor learning dynamics in both, adolescents and adults, and further shows that this effect is not solely driven by one group. Furthermore, our results remained consistent when including coupled spindle events in NREM2 (*Figure 3—figure supplement 2E*) and after outlier removal (*Figure 3—figure supplement 2F, G*).

To rule out age as a confounding factor that could drive the relationship between coupling strength, learning curve and task proficiency in the mixed sample, we used cluster-corrected partial correlations to confirm their independence of age differences (task proficiency: mean rho = 0.40, p = 0.017; learning curve: $rho_s$ = −0.47, p = 0.049). Additionally, given that we found that juggling performance could underlie a circadian modulation we controlled for individual differences in alertness

between subjects due to having just slept. We partialed out the mean PVT reaction time before the juggling performance test after sleep from the original analyses and found that our results remained unchanged (task proficiency: mean rho = 0.37, p = 0.025; learning curve: rho$_s$ = −0.49, p = 0.040). For a summary of the reported cluster-corrected partial correlations as well as analyses controlling for differences in sleep architecture, see *Figure 3—figure supplement 3*. Further, we also confirmed that our correlations are not influenced by individual differences in SO and spindle event parameters (*Figure 3—figure supplement 4*).

Finally, we investigated whether subjects with high coupling strength have a gross-motor learning advantage (i.e., trait effect) or a learning-induced enhancement of coupling strength is indicative for improved overnight memory change (i.e., state effect). First, we correlated SO–spindle coupling strength obtained from the adaptation night with the coupling strength in the learning night. We found that overall, coupling strength is highly correlated between the two measurements (mean rho across all channels = 0.55, *Figure 3—figure supplement 2H*), supporting the notion that coupling strength remains rather stable within the individual (i.e., trait). Second, we calculated the difference in coupling strength between the learning night and the adaptation night to investigate a possible state effect. We found no significant cluster-corrected correlations between coupling strength change and task proficiency—as well as learning curve change (*Figure 3—figure supplement 2I*).

Collectively, these results indicate the regionally specific SO–spindle coupling over central EEG sensors encompassing sensorimotor areas precisely indexes learning of a challenging motor task.

## Discussion

By comparing adolescents and adults learning a complex juggling task, we critically advance our previous work about the intricate interplay of learning and memory formation, brain maturation, and coupled sleep oscillations: First, we demonstrated that SO–spindle interplay precision is not only enhanced from childhood to late adolescence but also progressively improves from early adolescence to young adulthood (*Figure 3C*). Second and more importantly, we provide first evidence that the consistency of SO–spindle coordination is a promising model to track real-life gross-motor skill learning in addition to its key role in declarative learning (*Figure 3D, E*). Notably, this relationship between coupling and learning occurred in a regional specific manner and was pronounced over frontal areas for declarative and over motor regions for procedural learning (*Hahn et al., 2020*). Collectively, our results suggest that precise SO–spindle coupling supports gross-motor memory formation by integrating information from subcortical memory structures to cortical networks.

How do SO–spindle interactions subserve motor memory formation? Motor learning is a process relying on complex spatial and temporal scales in the human brain. To acquire motor skills the brain integrates information from extracortical structures with cortical structures via cortico-striato-thalamo-cortico loops and cortico-cerebello-thalamo-cortico circuits (*Dayan and Cohen, 2011*; *Doyon and Benali, 2005*; *Doyon et al., 2018*; *Pinsard et al., 2019*). However, growing evidence also advocates for hippocampal recruitment for motor learning, especially in the context of sleep-dependent memory consolidation (*Albouy et al., 2013*; *Boyke et al., 2008*; *Draganski et al., 2004*; *Pinsard et al., 2019*; *Sawangjit et al., 2018*; *Schapiro et al., 2019*). Hippocampal memory reactivation during sleep is one cornerstone of the active systems consolidation theory, where coordinated SO–spindle activity route subcortical information to the cortex for long-term storage (*Diekelmann and Born, 2010*; *Helfrich et al., 2019*; *Klinzing et al., 2019*; *Ngo et al., 2020*). Quantitative markers of spindle and SO activity but not the quality of their interaction have been frequently related to motor memory in the past (*Barakat et al., 2011*; *Bothe et al., 2019*; *Bothe et al., 2020*; *Huber et al., 2004*; *Morita et al., 2012*; *Nishida and Walker, 2007*; *Tamaki et al., 2008*). Our results now complement the active systems consolidation theories' mechanistic assumption of interacting oscillations by demonstrating that a precise SO–spindle interplay subserves gross-motor skill learning (*Figure 3D, E*). Of note, we did not derive direct hippocampal activity in the present study given spatial resolution of scalp EEG recordings. Nonetheless, as demonstrated recently, coupled spindles precisely capture corticohippocampal network communication as well as hippocampal ripple expression (*Helfrich et al., 2019*). Thus, higher SO–spindle coupling strength supporting gross-motor learning in our study points toward a more efficient information exchange between hippocampus and cortical areas.

Remarkably, hippocampal engagement is especially crucial at the earlier learning stages. Recently, it has been found that untrained motor sequences exhibit hippocampal activation that subsides for

more consolidated sequences. This change was further accompanied by increased motor cortex activation, suggesting a transformative function of sleep for motor memory (*Pinsard et al., 2019*). In other words, hippocampal disengagement likely indexes the transition from the fast learning stage to the slower learning stage with more proficient motor skill (*Dayan and Cohen, 2011*; *Doyon and Benali, 2005*). The dynamics of the two interacting learning stages of motor skill acquisition are likely reflected by the inverse relationship between task proficiency increases and learning curve attenuation (*Figure 2F*). Given that our subjects did not reach asymptotic performance level (*Figure 2A, B*) and that SO–spindle coupling tracks gross-motor skill learning dynamics as it relates to both, learning curve attenuation and task proficiency increments, it is plausible that SO-coupling strength represents the extent of hippocampal support for integrating information to motor cortices during complex motor skill learning.

Interestingly, SO and spindles are not only implicated in hippocampal–neocortical network communication but are also indicative for activity and information exchange in subcortical areas that are more traditionally related to the shift from fast to slow motor learning stages. For example, striatal network reactivation during sleep was found to be synchronized to sleep spindles, which predicted motor memory consolidation (*Fogel et al., 2017*). In primates, coherence between M1 and cerebellum in the SO and spindle frequency range suggested that coupled oscillatory activity conveys information through cortico-thalamo-cerebellar networks (*Xu et al., 2021*). One testable hypothesis for future research is whether SO–spindle coupling represents a more general gateway for the brain to exchange subcortical and cortical information and not just hippocampal–neocortical communication.

Critically, we found that the consistency of the SO–spindle interplay identified at electrodes overlapping with motor areas such as M1 was predictive for the gross-motor learning process (*Figure 3D, E*). This finding corroborates the idea that SO–spindle coupling supports the information flow between task-relevant subcortical and cortical areas. Recent evidence in the rodent model demonstrated that neural firing patterns in M1 during spindles became more coherent after performing a grasping motor task. The extent of neural firing precision was further mediated by a function of temporal proximity of spindles to SOs (*Silversmith et al., 2020*). Through this synchronizing process and their $Ca^{2+}$ influx propagating property, coupled spindles are likely to induce neural plasticity that benefits motor learning (*Niethard et al., 2018*).

How 'active' is sleep for real-life gross-motor memory consolidation? We found that sleep impacts the learning curve but did not affect task proficiency in comparison to a wake retention interval directly after learning (*Figure 2D, E*). Three accounts might explain the absence of a sleep effect on task proficiency. (1) Sleep rather stabilizes than improves gross-motor memory, which is in line with previous gross-motor adaption studies (*Bothe et al., 2019*; *Bothe et al., 2020*). This parallels findings in finger tapping tasks were the narrative evolved from sleep-related performance improvements (*Walker et al., 2002*) to stabilization (*Brawn et al., 2010*). (2) Presleep performance is critical for sleep to improve motor skills (*Wilhelm et al., 2012*). Participants commonly reach asymptotic presleep performance levels in finger tapping tasks, which is most frequently used to probe sleep effects on motor memory. Here, we found that using a complex juggling task, participants do not reach asymptotic ceiling performance levels in such a short time. Indeed, the learning progression for the *sleep-first* and *wake-first* groups followed a similar trend (*Figure 2A, B*), suggesting that more training and not in particular sleep drove performance gains. (3) Sleep effects are intermingled with time of day effects on juggling performance. Indeed, the steeper learning curve after the first retention interval in the sleep-first group can also be interpreted as a time of day effect. However, when modeling time of day and sleep specific effects across all performance blocks, we found a trend that sleep after learning supports task proficiency. Note, that the correlative nature of both factors in the model likely resulted in insufficient statistical power to produce independently significant results. Additionally, we did not find evidence for a circadian modulation of cognitive engagement based on objective reaction time data in our study (*Figure 2—figure supplement 1C*). However, a null-result does not exclude all possible circadian effects and ample evidence suggests that cognitive performance and motor learning are influenced by the time of day (*Blatter and Cajochen, 2007*; *Keisler et al., 2007*; *Tandoc et al., 2021*). Therefore, we cannot fully disentangle circadian and sleep effects with our study design, which should be considered a limitation to our findings.

Importantly, SO–spindle coupling still predicted learning dynamics on a single subject level advocating for a supportive function of sleep for gross-motor memory. Moreover, we found that SO–spindle

coupling strength remains remarkably stable between two nights, which also explains why a learning-induced change in coupling strength did not relate to behavior (*Figure 3—figure supplement 2I*). Thus, our results primarily suggest that strength of SO–spindle coupling correlates with the ability to learn (trait), but does not solely convey the recently learned information. Note that state and traits effects are not mutually exclusive. The overlap of state and trait effects is a long-standing issue in spindle literature, which also seems so apply to their coordinated interplay with SOs (*Lustenberger et al., 2015*; *Schabus et al., 2006*). This set of findings is in line with recent ideas that strong coupling indexes individuals with highly efficient subcortical–cortical network communication (*Helfrich et al., 2021*).

This subcortical–cortical network communication is likely to be refined throughout brain development, since we discovered elevated coupling strength in adults compared to early adolescents (*Figure 3C*). This result compliments our earlier findings of enhanced coupling precision from childhood to adolescence (*Hahn et al., 2020*) and the recently demonstrated lower coupling strength in preschool children (*Joechner et al., 2021*). We speculate that, similar to other spindle features, the trajectory of SO-coupling strength is likely to reach a plateau during adulthood (*Nicolas et al., 2001*; *Purcell et al., 2017*). Importantly, we identified similar methodological challenges to assess valid cross-frequency coupling estimates in the current cross-sectional study to the previous longitudinal study. Age severely influences fractal dynamics in the brain (*Figure 3—figure supplement 1A*) and the defining features of sleep oscillations (*Figure 3*, *Figure 3—figure supplement 1B*). Remarkably, interindividual oscillatory variability was pronounced even in the adult age group (*Figure 3A*), highlighting the critical need to employ individualized cross-frequency coupling analyses to avoid its pitfalls (*Aru et al., 2015*; *Muehlroth and Werkle-Bergner, 2020*).

Taken together, our results provide a mechanistic understanding of how the brain forms real-life gross-motor memory during sleep. However, how time of day additionally affects and interacts with sleep to support gross-motor learning remains an open question. As sleep has been shown to support fine-motor memory consolidation in individuals after stroke (*Gudberg and Johansen-Berg, 2015*; *Siengsukon and Boyd, 2008*), SO–spindle coupling integrity could be a valuable, easy to assess predictive index for rehabilitation success.

# Materials and methods

## Key resources table

| Reagent type (species) or resource | Designation | Source or reference | Identifiers | Additional information |
|---|---|---|---|---|
| Software, algorithm | Brain Vision Analyzer 2.2 | Brain Products GmbH https://www.brainproducts.com | RRID:SCR_002356 | |
| Software, algorithm | CircStat 2012 | *Berens, 2009* https://philippberens.wordpress.com/code/circstats/ | RRID:SCR_016651 | |
| Software, algorithm | EEGLAB 13_4_4b | *Delorme and Makeig, 2004* https://sccn.ucsd.edu/eeglab/index.php | RRID:SCR_007292 | |
| Software, algorithm | FieldTrip 20161016 | *Oostenveld et al., 2011* http://www.fieldtriptoolbox.org/ | RRID:SCR_004849 | |
| Software, algorithm | IRASA | *Wen and Liu, 2016* https://purr.purdue.edu/publications/1987/1 | | |
| Software, algorithm | MATLAB 2017a | MathWorks Inc | RRID:SCR_001622 | |
| Software, algorithm | RStudio | RStudio Team | RRID:SCR_000432 | |

*Continued on next page*

*Continued*

| Reagent type (species) or resource | Designation | Source or reference | Identifiers | Additional information |
|---|---|---|---|---|
| Software, algorithm | Somnolyzer 24 × 7 | Koninklijke Philips N.V. https://www.philips.co.in | | |
| Other | 'Jonglieren und Bewegungskünste' | *Sobota and Hollauf, 2013* Austrian ministry of Sports | | Juggling video instructions |

## Participants

We recruited 29 adolescents (mean ± SD age, 13.17 ± 0.85 years; 5 females, 24 males) from a local boarding school and 41 young adults (mean ± SD age, 22.24 ± 2.15 years; 16 females, 25 males) from the student population of the University of Salzburg. All participants were healthy, right-handed and without prior juggling experience. However, we excluded one adolescent for all analyses post hoc for violating the prior juggling experience criteria. Two adolescents did not participate in the third performance test. We randomly divided adolescents and adults into a *sleep-first* (adolescents: $N$ = 17, 12.94 ± 0.75 years; 3 females, 14 males; adults: $N$ = 25, 21.95 ± 2.42 years; 8 females, 17 males) and a *wake-first* group (adolescents: $N$ = 11, 13.36 ± 0.81 years; 2 females, 9 males; adults: $N$ = 16, 22.69 ± 1.62 years; 8 females, 8 males). See experimental design for more detailed information about the groups. We recorded PSG during full night sleep for all participants except adults in the *wake-first* group. Therefore, comparison of electrophysiological data between adults and adolescents was based on the adult *sleep-first* group and both adolescent groups. To ensure similar juggling learning experience, we only included adults and adolescents in the *sleep-first* group when analyzing the relationship between electrophysiological measures and behavioral performance. All participants and the legal custodians of the adolescents provided written informed consent before participating in the study. The study protocol was conducted in accordance with the Declaration of Helsinki and approved by the ethics committee of the University of Salzburg (EK-GZ:16/2014). Adults received monetary compensation or student credit for their participation. Adolescents received a set of juggling balls.

## Experimental design

Adults in the *sleep-first* group visited the sleep laboratory on three occasions (*Figure 1*). At the first day subjects slept in the sleep lab with full night PSG for adaptation purposes. On the second visit, subjects learned and practiced juggling by video instructions in the evening (8.45 pm to 9.45 pm). Juggling performance was assessed three times in total. The first performance test was conducted after the training session (10.00 pm to 10.18 pm). The second performance test (7.30 am to 7.48 am) took place after the first retention interval containing a full night of sleep with PSG (11 pm to 7 am). The third and last performance tests were executed after the second retention interval (9.00 pm to 9.18 pm) containing wakefulness. Adults in the *wake-first* group followed a similar protocol but with reversed order of the retention intervals (i.e., first retention interval containing wakefulness and the second interval containing sleep). Therefore, participants performed the juggling training (10.15 am to 11.15 am) and the first performance test (11.30 am to 11.48 am) in the morning, the second performance test after wakefulness (9.00 pm to 9.18 pm), and the third performance test after sleep (11.00 am to 11.18 am). We did not record PSG in the *wake-first* group because participants slept at home. To objectively assess attentiveness and potential circadian influences, all participants completed a psychomotor vigilance task (*Dinges and Powell, 1985*) before the performance tests. Actigraphy (Cambridge Neurotechnology Actiwatch, Cambridge, UK) and a sleep log (*Saletu et al., 1987*) verified compliance with a regular sleep schedule throughout the study.

Adolescents went through a study protocol comparable to the adults. However, we adjusted the protocol to adhere to the schedule of the boarding school and to control the training load. First, we recorded ambulatory PSG for both groups in their habitual sleep environment at the boarding school and second, we reduced the number of juggling blocks during the performance tests (for details see gross-motor task) because the study regime was already exhausting for our adult participants and we wanted to avoid a too excessive training load. The *sleep-first* group performed the juggling training (6.30 pm to 7.30 pm) and performance test in the evening (7.45 pm to 7.58 pm) followed by a retention interval containing sleep (21.00 pm to 6.00 am). The second performance test was conducted

after sleep (7.30 am to 7.43 am) and the third performance test after wakefulness (7.30 pm to 7.43 pm). The *wake-first* group learned to juggle (7.30 am to 8.30 am) with a subsequent performance test (8.45 am to 8.58 am) in the morning. The second performance test was executed after wakefulness in the evening (7.30 pm to 7.43 pm) and the third performance test was completed after sleep (7.30 am to 7.43 am).

## Gross-motor task

To investigate the involvement of SO–spindle coupling in acquiring a real-life gross-motor skill, we implemented a juggling paradigm, which has been shown to induce neural plasticity (*Boyke et al., 2008*; *Draganski et al., 2004*) and to be sensitive for sleep-dependent memory consolidation (*Morita et al., 2012*; *Morita et al., 2016*). Adults and adolescents completed the same juggling training, which was based on short video clips from the 'Juggling and Movement Arts' DVD ('Jonglieren und Bewegungskünste'; *Sobota and Hollauf, 2013*) containing step-by-step instructions from the correct stance to a full five-ball cascade (i.e., five continuous catches). We used 14 video clips demonstrating the exercises followed by a practice opportunity for the participants. The training session lasted approximately 1 hr with a short break after half an hour. During the performance tests, participants were instructed to juggle as accurately and continuously as possible. Adults juggled for five blocks a 3 min, which was always separated by a 30-s break. To alleviate the physical strain, adolescents only juggled for three blocks a 3 min during the performance tests. Training and performance tests were videotaped to evaluate the juggling performance.

## Parameterizing juggling performance

We evaluated the juggling performance by counting consecutive catches based on the video material. We used the number of three-ball cascades (i.e., three catches in a row, *Figure 2A, B*) as index for juggling performance by dividing the number of consecutive catches by three. We opted for three-ball cascades as a performance index because we considered three consecutive catches as the criteria for the motor task to qualify as juggling (*Boyke et al., 2008*; *Draganski et al., 2004*). Because juggling is a complex motor task where it is unlikely to reach ceiling level performance, we were interested in the progression of the learning process and how it is influenced by task proficiency. Therefore, we calculated a first-degree polynomial fit using the least-squares method to parameterize the learning curve ($m$, slope) per performance test block (*Figure 2A, B*, black lines and *Figure 2C, D*), using the formula:

$$m = \frac{\sum_{i=1}^{n} \left( x_i - \acute{X} \right) * \left( y_i - \acute{Y} \right)}{\sum_{i=1}^{n} \left( x_i - \acute{X} \right)}$$

Next, we calculated the intercept $c$ according to the following formula:

$$c = \acute{Y} - m * \acute{X}$$

Finally, task proficiency ($y_1$, *Figure 2E*) was estimated at the first time point of each performance test as

$$y_1 = m + c$$

## PSG and sleep staging

We recorded PSG with two systems. We conducted the ambulatory sleep recordings of the adolescents with a portable amplifier system (Alphatrace, Becker Meditec, Karlsruhe, Germany) with a sampling rate of 512 Hz. For in lab recordings of the adult participants, we utilized a 32-channel Neuroscan amplifier system (Scan 4.3.3 Software, Neuroscan Inc, Charlotte, NC) with a sampling rate of 500 Hz. Electrode placement was identical between the two recording systems and in accordance with the 10–20 system. Signals were recorded with gold cup electrodes placed at F3, Fz, F4, C3, Cz, C4, P3, Pz, P4, O1, and O2 on the scalp, as well as at A1 and A2 placed at the mastoids. To allow for sleep staging and to control for muscle artifacts, we recorded an electromyogram (bipolar electrodes at the musculus mentalis), a horizontal electrooculogram (EOG, above the right outer canthus and below the left outer canthus) and a vertical EOG (above and below the left eye). We used Cz as online reference and AFz as ground electrode. For sleep staging, we re-referenced the signal offline against contralateral mastoids. Sleep was semi-automatically staged in 30-s epochs

using the Somnolyzer 24 × 7 algorithm (Koninklijke Philips N.V.; Eindhoven, The Netherlands) and subsequently controlled by an expert scorer according to standard sleep staging criteria (*Iber et al., 2007*). For all other data analyses, we demeaned and re-referenced the EEG signal to a common average.

## Individualized cross-frequency coupling

To assess the precise interplay between SO and spindles, we used the same individualized cross-frequency coupling pipeline we developed earlier in order to account for network changes induced by aging, that are known to cause spurious coupling estimates (*Aru et al., 2015*; *Cole and Voytek, 2017*; *Hahn et al., 2020*; *Scheffer-Teixeira and Tort, 2016*). In brief, our approach was based on the following principles: (1) establishing the presence of sleep oscillations, (2) individually detecting transient oscillatory events, (3) alleviating power differences, and (4) ensuring co-occurrence of SO (phase providing signal) and sleep spindles (amplitude providing signal).

## Establishing sleep oscillations

First, we *z*-normalized the EEG signal in the time domain to mitigate prominent power differences and computed averaged power spectra from 0.1 to 30 Hz using a Fast Fourier Transform (FFT) routine with a Hanning window on 15 s of continuous NREM sleep (i.e., NREM2 and NREM3, *Figure 3—figure supplement 1A*, left) with a 1-s sliding window. Data are presented in the semi-log space. Next, we sought to isolate the oscillatory activity in the normalized data by means of irregular autospectral analysis (IRASA, *Wen and Liu, 2016*). We first derived the 1/f fractal component (*Figure 3—figure supplement 1A*, middle) from 15 s NREM sleep data in 1-s sliding steps and subsequently subtracted it from the power spectrum (*Figure 3—figure supplement 1A*, left) to obtain an unbiased estimate of the oscillatory activity for every subject on every electrode (*Figure 3—figure supplement 1A*, right and *Figure 3A*). To separate the 1/f component from the power spectrum, we used the same parameters as specified previously (*Hahn et al., 2020*). In short, the signal is stretched and compressed by the same noninteger factor (e.g., stretching by a factor of 1.1 and compressing by a factor of 0.9). We repeated the resampling with factors from 1.1 to 1.9 in 0.05 steps. This pair wise stretching and compressing systematically causes frequency peak shifts in the regular oscillatory activity but leaves the more random 1/f background activity unaffected. Because the oscillatory activity becomes faster by a similar factor as it becomes slower, the oscillatory activity is averaged out by median averaging across all pair wise resampled segments thus extracting the 1/f component. We then detected individual SO (<2 Hz) and spindle peak frequencies (10–17 Hz, *Figure 3—figure supplement 1B*) and the corresponding 1/f corrected amplitude (*Figure 3A*, left) in the oscillatory residual (*Figure 3—figure supplement 1A*, right). We considered the highest peak within the specified SO and spindle frequency ranges above as the most representative oscillatory event in each electrode. We then utilized the individual frequency peaks to inform the algorithms for discrete SO and spindle event detection.

## Individually detecting transient oscillatory events

We employed widely used spindle and SO detection algorithms (*Helfrich et al., 2018*; *Mölle et al., 2011*; *Staresina et al., 2015*) and adjusted them according to the 1/f corrected SO and spindle features for a fully individualized event detection (*Hahn et al., 2020*).

We detected spindle events (*Figure 3*, *Figure 3—figure supplement 1E*) by band-pass filtering the continuous signal ±2 Hz around the individual spindle peak per electrode. After filtering, we computed the instantaneous amplitude via a Hilbert transform. Next, we smoothed the signal with a running average in a 200-ms window. A sleep spindle was detected, when the signal exceeded the 75-percentile amplitude criterion for a time span of 0.5–3 s. We segmented the raw data ±2.5 s centered on the positive spindle peak.

We detected SO events (*Figure 3—figure supplement 1F*) by first high-pass filtering the continuous EEG signal at 0.16 Hz and then low-pass filtering at 2 Hz. Based on the filtered signal, we detected the zero-crossings that fulfilled the time criterion (length 0.8–2 s). The signal between two consecutive zero-crossings was considered a valid SO if its amplitude exceeded the 75-percentile threshold. We then segmented the raw data ±2.5 s centered on the negative peak.

## Alleviating power differences

Power differences in the signal can systematically impact cross-frequency coupling measures by changing the signal-to-noise ratio, which in turn influences the precision of the phase estimation of the signal (*Aru et al., 2015*; *Scheffer-Teixeira and Tort, 2016*). Because power decreases are apparent across the lifespan (*Campbell and Feinberg, 2009*; *Campbell and Feinberg, 2016*; *Hahn et al., 2020*; *Helfrich et al., 2018*), we *z*-normalized all detected SO and spindle events in the time domain to alleviate this possible confound before calculating phase-amplitude coupling measures (*Figure 3B*).

## Ensuring co-occurrence of SO and sleep spindles

Cross-frequency coupling renders meaningful information of network communication only when the suspected interacting oscillations are present in the signal. Therefore, we only analyzed SO and sleep spindle epochs during which they co-occurred in a 2.5-s time window (±~2 SO cycles around the spindle peak). Furthermore, we restricted all our coupling analyses to sleep stage NREM3 because of general lower co-occurrence of SO and spindles in NREM2 (*Figure 3—figure supplement 1C, D*), which can cause spurious coupling estimates (*Hahn et al., 2020*).

## Event-locked cross-frequency coupling

To parameterize the timed coordination between sleep spindles and SO (*Figure 3C*), we computed event-locked cross-frequency coupling analyses (*Dvorak and Fenton, 2014*; *Hahn et al., 2020*; *Helfrich et al., 2019*; *Helfrich et al., 2018*; *Staresina et al., 2015*) based on individualized and normalized spindle peak-locked segments. In short, we used a low-pass filter of 2 Hz to extract the underlying SO component (*Figure 3B*, inset) from the EEG signal and read out the phase angle corresponding with the sleep spindle peak after applying a Hilbert transform. We then calculated the coupling strength, which is defined as 1 − circular variance using the CircStat Toolbox function circ_r (*Berens, 2009*) to assess the consistency of the SO–sleep spindle interplay.

## Time–frequency analyses

We computed event-locked time–frequency representations based on −2 to 2 s epochs centered on the negative SO peak (*Figure 3—figure supplement 1F*). We used a 500-ms Hanning window in 50-ms steps to analyze the frequency power from 5 to 30 Hz in steps of 0.5 Hz. We subsequently baseline corrected the time–frequency representations by *z*-scoring the data based on the means and SDs of a bootstrapped distribution (10,000 iterations) for the –2- to −1.5-s time interval of all trials (*Flinker et al., 2015*; *Helfrich et al., 2018*).

## Statistical analyses

To compare juggling performance between the *sleep-first* and *wake-first* group and to assess the learning progression, we computed mixed ANOVAs with the between factor condition group (*sleep-first*, *wake-first*) and the repeated measure factor juggling blocks. Because number of juggling blocks differed between adolescents (9, *Figure 2A*) and adults (15, *Figure 2B*), we analyzed the juggling performance separately per age group. Influence of sleep on learning curve (*Figure 2D*) and task proficiency (*Figure 2E*) was assessed by a mixed ANOVA with the between factors condition group (*sleep-first*, *wake-first*) and age group (adolescents, adults) and the repeated factor performance test (preretention interval 1, postretention interval 1). To correct for multiple comparisons we clustered the data in the frequency (*Figure 3—figure supplement 1A*), time (*Figure 3B*), and space domain (*Figure 3*, *Figure 3—figure supplement 1B*), using cluster-based random permutation testing (Monte-Carlo method, cluster alpha 0.05, max size criterion, 1000 iterations, critical alpha level 0.05 two-sided; *Maris and Oostenveld, 2007*). Given our sparse sampling of only 11 scalp electrodes, we set the minimum number of neighborhood electrodes required to be included in the clustering algorithm to zero. For correlational analyses we utilized Spearman rank correlations (rho$_s$; *Figure 2F* and *Figure 3D, E*) to mitigate the impact of possible outliers as well as cluster-corrected Spearman rank correlations by transforming the correlation coefficients to *t*-values (p < 0.05) and clustering in the space domain (*Figure 3D, E*). Linear trend lines were calculated using robust regression. To control for possible confounding factors we computed cluster-corrected partial rank correlations (*Figure 3—figure supplements 3 and 4*). We report partial eta squared ($\eta^2$), Cohen's *d* (*d*) and

averaged Spearman correlation coefficients (mean rho) as effect sizes. Cluster effect sizes are estimated by first calculating Cohen's *d* for every data point in the significant cluster and subsequently averaging across the obtained values.

### Data analyses

We used functions from the Fieldtrip toolbox (*Oostenveld et al., 2011*), EEGlab toolbox (*Delorme and Makeig, 2004*), CircStat toolbox (*Berens, 2009*), and custom written code implemented in MatLab 2015a (Mathworks Inc) for data analyses. IRASA (*Wen and Liu, 2016*) was conducted using code obtained from the original research paper.

## Acknowledgements

This research was supported by Austrian Science Fund (P25000-B24) and the Centre for Cognitive Neuroscience Salzburg (CCNS). MAH was additionally supported by the Doctoral College 'Imaging the Mind' (FWF, Austrian Science Fund W1233-G17). RFH is supported by the German Research Foundation (DFG, HE 8329/2-1), the Hertie Foundation (Hertie Network of Excellence in Clinical Neurosciences), and the Jung Foundation for Science and Research (Ernst Jung Career Advancement Award).

## Additional information

### Funding

| Funder | Grant reference number | Author |
| --- | --- | --- |
| Austrian Science Fund | W1233-G17 | Michael A Hahn |
| Austrian Science Fund | P25000-B24 | Kerstin Hoedlmoser |
| Deutsche Forschungsgemeinschaft | HE 8329/2-1 | Randolph F Helfrich |
| Hertie Network of Excellence in Clinical Neuroscience | | Randolph F Helfrich |
| Jung Foundation for Science and Research | Ernst Jung Career Advancement Award | Randolph F Helfrich |

The funders had no role in study design, data collection, and interpretation, or the decision to submit the work for publication.

### Author contributions

Michael A Hahn, Conceptualization, Data curation, Formal analysis, Investigation, Methodology, Software, Validation, Visualization, Writing – original draft, Writing – review and editing; Kathrin Bothe, Investigation, Writing – review and editing; Dominik Heib, Data curation, Visualization, Writing – review and editing; Manuel Schabus, Methodology, Resources, Writing – review and editing; Randolph F Helfrich, Conceptualization, Methodology, Software, Supervision, Validation, Writing – review and editing; Kerstin Hoedlmoser, Conceptualization, Data curation, Funding acquisition, Investigation, Methodology, Project administration, Resources, Supervision, Validation, Writing – review and editing

### Author ORCIDs

Michael A Hahn ![orcid] http://orcid.org/0000-0002-3022-0552
Manuel Schabus ![orcid] http://orcid.org/0000-0001-5899-8772
Randolph F Helfrich ![orcid] http://orcid.org/0000-0001-8045-3111
Kerstin Hoedlmoser ![orcid] http://orcid.org/0000-0001-5177-4389

### Ethics

The study protocol was conducted in accordance with the Declaration of Helsinki and approved by the ethics committee of the University of Salzburg (EK-435 GZ:16/2014). Participants and their legal custodian provided written informed consent before entering the study.

Decision letter and Author response
Decision letter https://doi.org/10.7554/eLife.66761.sa1
Author response https://doi.org/10.7554/eLife.66761.sa2

## Additional files

### Supplementary files
- Transparent reporting form
- Supplementary file 1. Supplemental statistical data, analyses and sleep architecture.

### Data availability
The behavioral and electrophysiological preprocessed data and scripts to replicate the main conclusions and figures of the paper are available at https://doi.org/10.5061/dryad.qfttdz0gh.

The following dataset was generated:

| Author(s) | Year | Dataset title | Dataset URL | Database and Identifier |
|---|---|---|---|---|
| Hahn MA, Bothe K, Heib DPJ, Schabus M, Helfrich RF, Hoedlmoser K | 2021 | Slow oscillation-spindle coupling strength predicts real-life gross-motor learning in adolescents and adults | https://doi.org/10.5061/dryad.qfttdz0gh | Dryad Digital Repository, 10.5061/dryad.qfttdz0gh |

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
