## [Editor Report]

The authors used a clever design, in which adolescents and adults learned to juggle, to study the impact of sleep and associated oscillations on the consolidation of motor memory across age groups. Overall, the topic and the results of the present study are interesting and timely, and extends previous findings in the declarative memory domain to the motor memory domain.

---

## [Decision Letter]

**Decision letter after peer review:**

Thank you for submitting your article "Slow oscillation-spindle coupling strength predicts real-life gross-motor learning in adolescents and adults" for consideration by *eLife*. Your article has been reviewed by 2 peer reviewers, and the evaluation has been overseen by a Reviewing Editor and Laura Colgin as the Senior Editor. The reviewers have opted to remain anonymous.

Overall, the topic and the results of the present study are interesting and timely, and we appreciate the use of a more ecologically valid paradigm. However, several aspects of the analyses need further clarification, and critically, a question was raised as to whether this is a sleep story or a circadian rhythm story.

Essential revisions:

1. The results may first and foremost tell a circadian (rather than sleep) story. Examining the data in Figure 2A and 2B, it appears that every AM learning period has a higher learning curve (slope) than every PM period. While this could, of course, be due to having just slept, the main story gleaned from such a result is not a sleep effect on retention, which has been the emphasis in motor memory consolidation research in the last couple of decades, but on new learning. The fact that this effect appears present in the first session (juggling blocks 1-3 in adolescents and blocks 1-5 in adults) makes this seem the more likely story here, since it has less to do with "preparing one to re-learn" and more to do with just learning and when that learning is optimal. But even if it does not reach statistical significance in the first session alone, it remains a concern and should be considered a focus in the manuscript unless the authors can devise a reason to definitively rule it out. The authors should include all sessions from all subjects into a mixed effect model, predicting the slope of the learning curve with time of day and age group as fixed effects and subjects as random effects:

learning curve slope ~ AM/PM [AM (0) or PM (1)] + age [adolescent (0) or adult (1)] + (1|subject)

…or something similar with other regressors of interest. If this is significant for AM/PM status, they should re-try the analysis using only the first session. If this is significant, then a sleep-centric story cannot be defended here.

2. Related: The sleep data of all participants (thus from both sleep first and wake first) were used to determine the features of SO-spindle coupling in adolescents and adults. Were there any differences between groups (sleep first vs. wake first)? This might be interesting in general but especially because only data of the sleep first group entered the subsequent correlational analyses.

3a. Supporting and extending previous work of the authors (Hahn et al., 2020), SO-spindle coupling over centro-parietal areas was stronger in adults as compared to adolescents. Despite these differences in the EEG results the authors collapsed the data of adults and adolescents for their correlational analyses (Figure 4a and 4b). Why would the authors think that this procedure is viable (also given the fact that different EEG systems were used to record the data)?

3b. If the authors believe it is justified to combine these groups, Figure 3 and 4 should be combined and some current figure panels in Figure 3 should be removed or moved to the supplementary information.

4. The authors might want to explicitly show that the reported correlations (with regards to both learning curve and task proficiency change) are not driven by any outliers. It would be useful to know if the relationship is significant with Pearson correlations when robust regression is applied.

5. With only a single night of recording data, it is impossible to disentangle possible trait-based sleep characteristics (e.g., Subject 1 has high SO-spindle coupling in general and retains motor memories well, but these are independent of each other) from a specific, state-based account (e.g., Subject 1's high SO-spindle coupling on night 1 specifically led to their improved retention or change in learning, etc., and this is unrelated to their general SO-spindle coupling or motor performance abilities). Clearly, many studies face this limitation, but this should be acknowledged.

6. The authors used a partial correlation analysis to rule out that age drove the relationship between coupling strength, learning curve and task proficiency. It seems like this analysis was done specifically for electrode C4, after having already established that coupling strength at electrode C4 correlates in general with changes in the learning curve and task proficiency. The claim that results were not driven by age as confounding factor would be stronger if the authors used a cluster-corrected partial correlation in the first place (just as in the main analysis).

7. To allow a more comprehensive assessment of the underlying data information with regards to general sleep descriptives (minutes, per cent of time spent in different sleep stages, overall sleep time etc.) as well as related to SOs, spindles and coupled events (e.g. number, density etc.) would be needed.

8. The authors state that "To ensure the simultaneous presence of the two interacting sleep oscillations in the signal, we restricted our analyses to NREM3 sleep given the higher co-occurrence rate." We do not understand this reasoning. The utilized procedure of specifically isolating sleep spindles that are followed or preceded by slow oscillations already ensures the presence of SOs and sleep spindles in the data. Hence, why not take coupled events from sleep stage N2 into account? Or do the authors think that light sleep SO-spindle events are qualitatively different from SWS SO-spindle complexes (and if so does the present data support such a notion)?

[Editors’ note: further revisions were suggested prior to acceptance, as described below.]

Thank you for resubmitting your work entitled "Slow oscillation-spindle coupling strength predicts real-life gross-motor learning in adolescents and adults" for further consideration by *eLife*. Your revised article has been evaluated by Laura Colgin (Senior Editor) and a Reviewing Editor.

The manuscript has been improved but there are some remaining issues that need to be addressed, as outlined below:

The authors have done an impressive job with this revision. It is meticulously organized, thorough, and clearly stated. That is all to their major credit. However, I still cannot come to agree that their data supports much of the story they are telling.

First, part of the issue may be the change from their original story and the new one following the revision. Making major revisions can obviously be tricky, especially when a revision requires as many changes as theirs did (and I again commend them on the overhaul). But there is still something unclear in their primary claims. They say in their cover letter, "Collectively, our results suggest that SO-spindle coupling indexes the integrity of memory pathways (as reviewed in detail recently: Helfrich et al., 2021); thus, reflects a trait-specific (in contrast to a state-specific) correlate of learning capacity." However, this story does not come through clearly in the new paper. In fact, reading the new paper, it seems this is nodded to only here in the Discussion – "Thus, our results primarily suggest that strength of SO-spindle coupling correlates with the ability to learn (trait), but does not solely convey 534 the recently learned information. This set of findings is in line with recent ideas that strong coupling indexes individuals with highly efficient subcortical-cortical network communication (Helfrich et al., 2021)." Much of the paper instead talks about active systems consolidation theory, which I believe is not supported in their data, and the authors do seem to agree. If the authors indeed want to make this more of a memory pathway integrity story, it seems more unpacking of the ideas in their recent review is warranted, as does perhaps some evidence in the literature linking sleep measures to integrity in some neural pathways (e.g., Mander et al., 2017).

Second, they concede in various locations that the circadian story cannot be ruled out, which I also commend, but then the paper still largely revolves around active sleep consolidation theory. I invite the authors to imagine convincing a hypothetical researcher who thought the brain just shuts off entirely during sleep ("sleep does nothing") and that people have different abilities based on the time of day. (Believe it or not, this is not my belief.) How would the authors convince this person based on these behavioral data that sleep is actually doing something? I do not know whether they could, given the mixed-effects model findings.

Of course, they could point to the prior literature. The prior literature on sleep and motor learning has shown, in the case of the Morita juggling studies cited, that there should be better overall performance after sleep (vs equivalent wake periods). And in the case of countless finger-tapping studies, even though the major story has changed from one of absolute improvement (e.g., Walker et al., 2002) to stabilization (e.g., Brawn et al., 2010) after sleep, there seem to be sleep (vs wake) benefits on overall performance (analogous to task proficiency here). This, however, is not what the authors find with their learning curve findings here, as performance seems, if anything, worse on the first few trials after sleep (though this may not be significant) and then catches up more quickly. So, it is hard to know whether the prior literature would necessarily help them convince this researcher about their own findings.

This researcher may also say that the inclusion of a PVT is great, and the null results across sessions is more helpful than not to their story. But this researcher may add that a null PVT difference does not exclude all possible circadian effects. There are certainly circadian effects on cognition – including the very recent publication of Tandoc et al., (2021) and even on motor learning (Keisler et al., 2007) – and indeed the authors do find such an effect here in their mixed-effects model analysis. Therefore, the null PVT results are not conclusive, especially in counteracting an effect that they actually found in their paper.

One could then point this researcher to the SO-spindle coupling results as evidence that sleep is playing a strong role here. However, given that these are trait- vs. state-based results, it is unclear why stronger SO-spindle coupling for some individuals – which may be having an impact on neural integrity over a long timescale – would prime their nervous systems for more learning right after sleep than at some other time during the day. The researcher may say, okay, SO-spindle coupling results do not prove sleep does anything, they merely correlate with the observed behavioral result, and moreover, they constitute a trait (vs post-learning sleep state) effect. They may add that it is also unclear why, if stronger SO-spindle coupling is doing something, it could not alternatively reflect some other individual trait that could even lead to the observed circadian effects that learning curves are higher in the morning.

One analysis that could possibly work to disentangle circadian vs. active sleep effects would be to include a different factor in the mixed-effects model that could tease apart time of day from sleep-after-learning effects. In addition to including “Time of day”, where all mornings = 1 and all evenings = 0, the authors could include the conjunction of “Time of day + after learning”, where mornings on the 2^nd^ and 3^rd^ sessions = 1 and mornings on the 1^st^ session and all evenings = 0. This would capture the idea that post-learning mornings show differential improvement because post-learning sleep sort of “prepared” the networks to re-learn within a short time span, and this preparation was not operative before the 1^st^ session. I say it could “possibly” work above because the two factors would still be quite correlated (identical except for the first morning session), which could hurt their statistical power to independently produce effects. Nevertheless, if BOTH factors end up being significant, I think the authors could make the claim that both are contributing (that is, time of day + after learning is actually independently contributing above and beyond what time of day could do alone). If only one is significant, then the story is clean, but may have to change. If neither are significant, then it may be difficult to know what to do, and the authors may have to fall back on the original time-of-day analysis and keep things closer to as is but acknowledge more of the uncertainty surrounding the effects. If nothing changes upon a second revision in this regard, I do expect the authors to incorporate circadian possibilities more thoroughly in their paper, such as in their abstract and with more citations of this literature.

I realize this may seem a lot for a second round of revisions, and the authors have clearly done an impressive amount of work on the paper, but I feel that the authors can still strengthen it, either with this last analysis or by refocusing on the stories that can and cannot be supported here. There is something here that lacks clarity in translating from the data to the story about them, and, as a result, it remains difficult to confidently find the main takeaway from the manuscript.

---

## [Author Response]

Essential revisions:1. The results may first and foremost tell a circadian (rather than sleep) story. Examining the data in Figure 2A and 2B, it appears that every AM learning period has a higher learning curve (slope) than every PM period. While this could, of course, be due to having just slept, the main story gleaned from such a result is not a sleep effect on retention, which has been the emphasis in motor memory consolidation research in the last couple of decades, but on new learning. The fact that this effect appears present in the first session (juggling blocks 1-3 in adolescents and blocks 1-5 in adults) makes this seem the more likely story here, since it has less to do with “preparing one to re-learn” and more to do with just learning and when that learning is optimal. But even if it does not reach statistical significance in the first session alone, it remains a concern and should be considered a focus in the manuscript unless the authors can devise a reason to definitively rule it out. The authors should include all sessions from all subjects into a mixed effect model, predicting the slope of the learning curve with time of day and age group as fixed effects and subjects as random effects:learning curve slope ~ AM/PM [AM (0) or PM (1)] + age [adolescent (0) or adult (1)] + (1|subject)…or something similar with other regressors of interest. If this is significant for AM/PM status, they should re-try the analysis using only the first session. If this is significant, then a sleep-centric story cannot be defended here.

We thank the reviewer for the insightful comment and the detailed suggestion for how to further capture the temporal dynamics of learning in the juggling task.

The reviewer raises an important issue, which often pertains studies that examine sleep-dependent memory formation and it is inherent to our task design that sleep (state-specific) and circadian effects cannot fully be disentangled.

Therefore, we actually performed a control task in order to circumvent this issue and to assess the overall cognitive engagement as a function of time of day. Note, we did not report this data in the initial submission, but it is now included (see below). Furthermore, we also conducted the suggested linear mixed model analyses to obtain an additional statistical metric to quantify the temporal dynamics of learning.

Importantly, we would like to re-emphasize that our key results primarily suggest that the strength of SO-spindle coupling correlates with the ability to learn, but does not solely convey the recently learned information.

Collectively, our results suggest that SO-spindle coupling indexes the integrity of memory pathways (as reviewed in detail recently: Helfrich et al., 2021); thus, reflects a trait-specific (in contrast to a state-specific) correlate of learning capacity. For the full statistical analyses and all results, please see our response to issue #5 below. In the following, we separately address the reviewers’ comments based on our (1) control analyses and (2) the additional linear mixed models.

1. Control analyses from an independent task

To assess whether juggling performance is impacted by circadian rhythmicity (i.e. influenced by time of day; differences in tiredness and alertness; sleep inertia), we analyzed the reaction times in a psychomotor-vigilance-task (PVT; Dinges and Powell, 1985). The PVT is a reaction time task that is considered to be the gold standard to assess alertness and vigilance due to its high sensitivity to sleep loss and circadian influences (Dinges et al., 1997; Killgore, 2010; Van Dongen et al., 2003). In our experiment, we conducted a PVT before each juggling performance test. Thus, if a circadian effect indeed confounded our analysis, we artid expect a significant interaction between condition groups (sleep first vs wake first) and performance test (pre vs post retention interval 1) in this objective reaction time task.

We found no significant difference between the groups (Figure 2 – figure supplement 1C A, F(1,67) = 1.87, p = 0.18, partial eta² = 0.03) nor between the performance tests (F(1,67) = 1.06, p = 0.31, partial eta² = 0.02). Critically, we found no significant interaction (F(1,67) = 0.35, p = 0.55, partial eta² = 0.01) indicating that participants’ cognitive engagement did not differ in the juggling performance tests due to the preceding sleep or wake intervals. Taken together, these results rule out a strong circadian influence on the juggling performance.

Next, we computed cluster-corrected partial correlations to control whether our initially reported correlations between coupling strength and task performance are influenced by individual differences in cognitive engagement due to having just slept. When controlling for mean PVT reaction time in the morning our results remained unchanged, showing that subjects with higher coupling strength have better task proficiency after sleep (mean rho = 0.37, p = 0.025) and flatter learning curves (rho = -0.47, p = 0.049). These control analyses as well as other cluster-corrected partial correlations for various possible confounding factors (see our answer to issue #7) are now fully reported in the revised version of the manuscript as Figure 3 —figure supplement 3.

Taken together these analyses show that first, we found no evidence for circadian rhythmicity as a confounding factor between the groups and second that our correlational analyses were not impacted by differences in individual cognitive engagement.

2. Linear mixed model analyses

2.1. Learning curve

We followed the reviewer’s suggestion and predicted the learning curve using a linear mixed model with age group and time of day (i.e. performance test in the morning or evening) as fixed effects and subjects as random effects (Learning curve ~ Age group + Time of day + (1|Subjects)). The following linear mixed models were computed with the fitlme.m matlab function using maximum likelihood estimation.

We found that the learning curve was flatter in the performance tests in the evening than in the morning (Β = -1.129, t(202) = -2.885, p = 0.004, CI_95_ = [-1.901, -0.357], for the full report see Table 5 in Supplementary file 1). Next, we predicted the learning curve only in the first performance test using an identically structured linear mixed model. As expected, the learning curve was flatter in the evening than in the morning (Β = -1.294, t(66) = -2.885, p = 0.028, CI_95_ = [-2.442, -0.146], for the full report see Table 5 in Supplementary file 1).

Combined, these results could in principle indicate a circadian modulation of the juggling learning process. However, one cannot reject the hypothesis that sleep might have an additional impact. We fully acknowledge this shortcoming and address it in detail in the discussion.

2.2. Task proficiency

Another possible explanation for the difference in learning curves between morning and evening is a systematic difference in task proficiency, given that we found that both parameters are inversely correlated (Figure 2F). This is in line with the notion that task performance determines the magnitude of subsequent performance gains (Dayan and Cohen, 2011).

Using an identically structured linear mixed model as above, we found that task proficiency was generally better in the evening performance tests (Β = 2.7467, t(202) = 0.951, p = 0.047, CI_95_ = [0.037, 5.456], for the full report see Table 5 in Supplementary file 1). However, as illustrated in Figure 2AB, it is likely that this effect is mainly driven by the adult sleep first group as initially reported in the mixed ANOVA (Figure 2E, F(1,65) = 5.210, p = 0.026, partial eta² = 0.074).

In summary, differences in learning curve depending on time of day are explained by differences in task proficiency (i.e. the better one already performs in a task the harder it is to improve). Even though it is impossible to completely rule out circadian influences in the current task-design, we could not find any evidence for a differences in alertness (Figure 2 – figure supplement 1C A). Further, while we do interpret our result based on the mechanistic assumption of the active systems consolidation theory of interacting sleep oscillations supporting memory formation, our results rather suggest a trait- than a state-effect (please refer to our response to issue #5 for more details). We also discuss this issue in the revised discussion.

2.3. PVT reaction time

Finally, we wanted to specifically probe whether we can find evidence for a circadian modulation using a higher statistically powered linear mixed model (as compared to the general linear model in Figure 2 – figure supplement 1C). Therefore, we also predicted the PVT reaction time with age group and time of day as fixed effects and allowed for random intercepts for subjects.

We found no significant difference in reaction times in the PVT between performance tests in the morning and in the evening (Β = -1.107, t(202) = -0.238, p = 0.812, CI_95_ = [-10.29, 8.077], for the full report see Table 6). Likewise, there were no differences in reaction time when only considering the reaction times for the first performance test (Β = 9.622, t(66) = 0.951, p = 0.345, CI_95_ = [-10.586, 29.831], for the full report see Table 6).

Together, this suggests that there were no differences in alertness and vigilance as a function of time of day in the performance tests. This observation further supports our initial control analysis (Figure 2 – figure supplement 1C). Notably, a simple reaction time task is not fully comparable to the complex juggling learning process, which is again addressed in the discussion. In summary, we did not find evidence for a circadian modulation in our sample based on the objective reaction time data.

Table note: we used reference dummy coding, where the coefficient of the first category is set to 0 (i.e. fixed effect of age group is referenced to adolescents whereas the Time of day fixed effect is referenced to performance tests in the morning).

3. Conclusion

Taken together, our behavioral results might be compatible with a sleep or circadian effect. The analysis from the control task does not indicate the presence of strong behavioral effects. Furthermore, it is in fact likely that sleep- and circadian-effects are simultaneously present. However, given that we observed robust electrophysiological correlates that indicate a trait- and not a state-like effect, we are convinced that a circadian effect does not compromise our results and the conclusions.

4. Changes in the manuscript

All of these important control analyses are now included in the revised version of the manuscript.

We now refer to these analyses in the Results section (page 9, lines 174 – 181):

“Figure 2A and 2B indicate that performance tests in the morning might be characterized by a steeper learning curve than the evening tests. We confirmed this observation using a linear mixed model (Supplementary file – table 5AB). While this finding might also indicate a circadian influence on learning in our task, we did not find evidence for an effect on circadian sensitive psychomotor vigilance task reaction time. Neither when comparing sleep first and wake first groups (Figure 2 —figure supplement 1C), nor when specifically probing evening and morning performance tests (Supplementary file – table 5EF).”

And (page 16, lines 351 – 360):

“Additionally, given that we found that juggling performance could underlie a circadian modulation we controlled for individual differences in alertness between subjects due to having just slept. We partialed out the mean PVT reaction time before the juggling performance test after sleep from the original analyses and found that our results remained unchanged (task proficiency: mean rho = 0.37, p = 0.025; learning curve: rhos = -0.49, p = 0.040). For a summary of the reported cluster-corrected partial correlations as well as analyses controlling for differences in sleep architecture see Figure 3 —figure supplement 3.”

In the light of these results, we now discuss the sleep effect on gross-motor learning and a potential circadian influence in detail in the Discussion section (page 22 – 23, lines 502 – 518):

“How relevant is sleep for real-life gross-motor memory consolidation? We found that sleep impacts the learning curve but did not affect task proficiency in comparison to a wake retention interval (Figure 2DE). […] Here we found that using a complex juggling task, participants do not reach asymptotic ceiling performance levels in such a short time. Indeed, the learning progression for the sleep-first and wake-first groups followed a similar trend (Figure 2AB), suggesting that more training and not in particular sleep drove performance gains. We note that juggling performance in our study could have been influenced by the timing of when learning is optimal in the circadian cycle. However, we did not find evidence for a circadian modulation of cognitive engagement based on objective reaction time data (Figure 2 —figure supplement 1C).”

We further clearly state that the possibility of simultaneously present sleep- and circadian effect is a limitation to our study (page 23, lines 518 – 519):

“Nonetheless, we cannot fully disentangle circadian and sleep effects with our study design.”

Additionally, we discuss that our correlations rather reflect a trait-effect (page 23, lines 521 – 528):

“Moreover, we found that SO-spindle coupling strength remains remarkably stable between two nights, which also explains why a learning-induced change in coupling strength did not relate to behavior (Figure 3 —figure supplement 2I). Thus, our results primarily suggest that strength of SO-spindle coupling correlates with the ability to learn (trait), but does not solely convey the recently learned information. This set of findings is in line with recent ideas that strong coupling indexes individuals with highly efficient subcortical-cortical network communication (Helfrich et al., 2021).”

2. Related: The sleep data of all participants (thus from both sleep first and wake first) were used to determine the features of SO-spindle coupling in adolescents and adults. Were there any differences between groups (sleep first vs. wake first)? This might be interesting in general but especially because only data of the sleep first group entered the subsequent correlational analyses.

We thank the reviewers for their remark. We agree that adding additional information about possible differences between the sleep first and wake first groups would allow for a more comprehensive assessment of the reported data. We did not explain our reasoning to include only the sleep first groups for the correlation analyses clearly enough in the original manuscript. Unfortunately, we can only report data for the adolescents in our sample, because we did not record polysomnography (PSG) for the adult wake first group. This is also one of the two reasons why we focused on the sleep first groups for our correlational analyses.

Adolescents in the sleep first group did not differ from adolescents in the wake first group in terms of sleep architecture (except REM (%), which did not correlate with behavior [task proficiency: rho = -0.17, p = 0.28; learning curve: -0.02, p = 0.90]) as well as SO and sleep spindle event descriptive measures (see Table 7 in Supplementary file 1). Importantly, we found no differences in coupling strength between the two groups (Figure 3 – figure supplement 2AB).

The second reason why we focused our analyses on sleep first was that adolescents in the wake first group had higher task proficiency after the sleep retention interval than the sleep first group (Figure 3 – figure supplement 2A; t(23) = -2.24, p = 0.034). This difference in performance is directly explained by the additional juggling test that the wake first group performed at the time point of their learning night, which should be considered as additional training. Therefore, we excluded the wake first group from our correlational analyses because sleep and wake first group are not comparable in terms of juggling training during the night when we assessed SO-spindle coupling strength.

These additional analyses and the summary statistics of sleep architecture and SO/spindle event descriptives of adolescents in the sleep first and wake first group, are now reported in the revised version of the manuscript as Figure 3 —figure supplement 2AB and Supplementary file – table 7.

We now explicitly explain our rationale of why we only considered participants in the sleep first group for our correlational analyses in the Results section (page 6, lines 101 – 105):

“Polysomnography (PSG) was recorded during an adaptation night and during the respective sleep retention interval (i.e. learning night) except for the adult wake-first group (for sleep architecture descriptive parameters of the adaptation night and learning night as well as for adolescents and adults see Supplementary file – table 1 and 2).”

And (page 15, lines 311 – 320):

“Furthermore, given that we only recorded polysomnography for the adults in the sleep first group and that adolescents in the wake first group showed enhanced task proficiency at the time point of the sleep retention interval due to additional training (Figure 3 —figure supplement 2A), we only considered adolescents and adults of the sleep-first group to ensure a similar level of juggling experience adolescents and adults of the sleep-first group to ensure a similar level of juggling experience (for summary statistics of sleep architecture and SO and spindle events of subjects that entered the correlational analyses see Supplementary file – table 6). Notably, we found no differences in electrophysiological parameters (i.e. coupling strength, event detection) between the adolescents of the wake first and sleep first group (Figure 3 —figure supplement 2B and Supplementary file – table 7).”

3a. Supporting and extending previous work of the authors (Hahn et al., 2020), SO-spindle coupling over centro-parietal areas was stronger in adults as compared to adolescents. Despite these differences in the EEG results the authors collapsed the data of adults and adolescents for their correlational analyses (Figure 4a and 4b). Why would the authors think that this procedure is viable (also given the fact that different EEG systems were used to record the data)?

We thank the reviewers for the opportunity to clarify why we think it is viable to collapse the data of adolescents and adults for our correlational analyses. In the following we split our answers based on the two points raised by the reviewers: (1) electrophysiological differences (i.e. coupling strength) between the groups and (2) potential signal differences due to different EEG systems.

1. Electrophysiological differences

Upon inspecting the original Figure 4, it is apparent that the coupling strength of the combined sample does not form isolated clusters for each age group. In other words, while adult coupling strength is on the higher and adolescent coupling on the lower end due to the developmental increase in coupling strength we reported in the original Figure 3F, both samples overlap forming a linear trend. Second, when running the correlational analyses between coupling strength and task proficiency as well as learning curve separately for each age group, we found that they follow the same direction. Adolescents with higher coupling strength show better task proficiency (rho_s_ = 0.66, p = 0.005). This effect was also present when using robust regression (b = 109.97, t(15)=3.13, rho = 0.63, p = 0.007). Like adolescents, adults with higher coupling strength at C4 displayed better task proficiency after sleep (rho_s_ = 0.39, p = 0.053). This relationship was stronger when using robust regression (b = 151.36, t(23)=3.17, rho = 0.56, p = 0.004). For learning curves, we found the expected negative correlation at C4 for adolescents (rho_s_ = -0.57, p = 0.020) and adults (rho_s_ = -0.44, p = 0.031). Results were comparable when using robust regression (adolescents: b = -59.58, t(15) = -2.94, rho = -0.60, p = 0.010; adults: b = -21.99, t(23 ) = -1.71, rho = -0.37, p = 0.101).

Taken together, these results demonstrate that adolescents and adults show the effects and the same direction at the same electrode, thus, making it highly unlikely that our results are just by chance and that our initial correlation analyses are just driven by one group.

Additionally, we already controlled for age in our original analyses using partial correlations (also refer to our answer to issue #6). Hence, our additional analyses provide additional support that it is viable to collapse the analyses across both age groups even though they differ in coupling strength.

2. Different EEG-systems

The reviewers also raise the question whether our analyses might be impacted by the different EEG systems we used to record our data. This is an important concern especially when considering that cross-frequency coupling analyses can be severely confounded by differences in signal properties (Aru et al., 2015). In our sample, the strongest impact factor on signal properties is most likely age, given the broadband power differences in the power spectrum we found between the groups (original Figure 3A). Importantly, we also found a similar systematic power difference in our longitudinal study using the same ambulatory EEG system for both data recordings (Hahn et al., 2020). This is in line with numerous other studies demonstrating age related EEG power changes in broadband- as well as SO and sleep spindle frequency ranges (Campbell and Feinberg, 2016; Feinberg and Campbell, 2013; Helfrich et al., 2018; Kurth et al., 2010; Muehlroth et al., 2019; Muehlroth and Werkle-Bergner, 2020; Purcell et al., 2017). Therefore, we already had to take differences in signal property into account for our cross-frequency analyses. Regardless whether the underlying cause is an age difference or different signal-to-noise ratios of different EEG systems.

To mitigate confounds in the signal, we used a data-driven and individualized approach detecting SO and sleep spindle events based on individualized frequency bands and a 75-percentile amplitude criterion relative to the underlying signal. Additionally we z-normalized all spindle events prior to the cross-frequency coupling analyses. We found no amplitude differences around the spindle peak (point of SO-phase readout) between adolescents that were recorded with an ambulatory amplifier system (alphatrace) and adults that were recorded with a stationary amplifier system (neuroscan) using cluster-based random permutation testing. This was also the case for the SO-filtered (< 2 Hz) signal. Critically, the significant differences in amplitude from -1.4 to -0.8 s (p = 0.023, d = -0.73) and 0.4 to 1.5 s (p < 0.001, d = 1.1) are not caused by age related differences in power or different EEG-systems but instead by the increased coupling strength (i.e. higher coupling precision of spindles to SOs) in adults giving rise to a more pronounced SO-wave shape when averaging across spindle peak locked epochs.

Consequently, our analysis pipeline already controlled for possible differences in signal property introduced through different amplifier systems. Nonetheless, we also wanted to directly compare the signal-to-noise ratio of the ambulatory and stationary amplifier systems. However, we only obtained data from both amplifier systems in the adult sleep first group, because we recorded EEG during the juggling learning phase with the ambulatory system in addition to the PSG with the stationary system. First, we computed the power spectra in the 1 to 49 Hz frequency range during the juggling learning phase (ambulatory) and during quiet wakefulness (stationary) for every subject in the adult sleep first group in 10-seconds segments. Next, we computed the signal-to-noise ratio (mean/standard deviation) of the power spectra per frequency across all segments. We only found a small negative cluster from 21.9 to 22.5 Hz (p = 0.042, d = 0.53), which did not pertain our frequency-bands of interest. Critically, the signal-to-noise ratio of both amplifiers converged in the upper frequency bands approaching the noise floor, therefore, strongly supporting the notion that both systems in fact provided highly comparable estimates.

In conclusion, both age groups display highly similar effects and direction when correlating coupling strength with behavior. Further, after individualization and normalization the analytical signal, we found no differences in signal properties that would confound the cross-frequency analysis. Lastly, we did not find systematic differences in signal-to-noise ratio between the different EEG-systems. Thus, we believe it is justified to collapse the data across all participants for the correlational analyses, as it combines both, the developmental aspect of enhanced coupling precision from adolescence to adulthood and the behavioral relevance for motor learning which we deem a critical research advance from our previous study.

We have now added Figure 3B to the revised version of the manuscript to demonstrate that there were no systematic differences between the two age groups in the analytical signal due to the expected age related power differences or EEG-systems.

Specifically, we now state in the Results section (page 13 – 14, lines 282 – 294):

“We assessed the cross frequency coupling based on z-normalized spindle epochs (Figure 3B) to alleviate potential power differences due to age (Figure 3 —figure supplement 1A) or different EEG-amplifier systems that could potentially confound our analyses (Aru et al., 2015). Importantly, we found no amplitude differences around the spindle peak (point of SO-phase readout) between adolescents and adults using cluster-based random permutation testing (Figure 3B), indicating an unbiased analytical signal. This was also the case for the SO-filtered (< 2 Hz) signal (Figure 3B, inset). Critically, the significant differences in amplitude from -1.4 to -0.8 s (p = 0.023, d = -0.73) and 0.4 to 1.5 s (p < 0.001, d = 1.1) are not caused by age related differences in power or different EEG-systems but instead by the increased coupling strength (i.e. higher coupling precision of spindles to SOs) in adults giving rise to a more pronounced SO-wave shape when averaging across spindle peak locked epochs.”

Further, we added the correlational analyses that we computed separately for the age groups to the revised manuscript (Figure 3 —figure supplement 2CD) as they further substantiate our claims about the relationship between SO-spindle coupling and gross-motor learning.

We now refer to these analyses in the Results section (page 16, lines 338 – 343):

Critically, when computing the correlational analyses separately for adolescents and adults, we identified highly similar effects at electrode C4 for task proficiency (Figure 3 —figure supplement 2C) and learning curve (Figure 3 —figure supplement 2D) in each group. These complementary results demonstrate that coupling strength predicts gross-motor learning dynamics in both, adolescents as well as adults, and further show that this effect is not solely driven by one group.

3b. If the authors believe it is justified to combine these groups, Figure 3 and 4 should be combined and some current figure panels in Figure 3 should be removed or moved to the supplementary information.

We thank the reviewers for their suggestion and we agree that the figures of our manuscript would benefit from more focus. Therefore, we combined Figure 3 and 4 from the original manuscript into a revised Figure 3 in the updated version of the manuscript. In more detail, subpanels that explain our methodological approach can now be found in Figure 3 —figure supplement 1, while the updated Figure 3 now focuses on developmental changes in oscillatory dynamics and SO-spindle coupling strength as well as their relationship to gross-motor learning.

4. The authors might want to explicitly show that the reported correlations (with regards to both learning curve and task proficiency change) are not driven by any outliers. It would be useful to know if the relationship is significant with Pearson correlations when robust regression is applied.

We thank the reviewers for their suggestion. We agree that when inspecting the scatter plots it looks like that the correlations could be severely influenced by two outliers in the adult group. Because this is an important matter, we recalculated all previously reported correlations without the two outliers (Author response image 1, left column) and followed the reviewer’s suggestion to also compute robust regression (Author response image 1, right column) and found no substantial deviation from our original results.

**Author response image 1. sa2fig1:** (A) Spearman rank correlation between task proficiency change and learning curve change collapsed across adolescents (red dot) and adults (black diamonds) after removing two outlier subjects in the adult age group. Grey-shaded area indicates 95% confidence intervals of the robust trend line. (B) Robust regression of task proficiency change and learning curve change of the original sample. (C) Cluster-corrected correlations (right) between individual coupling strength and overnight task proficiency change (post – pre retention) after outlier removal (left, spearman correlation at C4, uncorrected). Asterisks indicate cluster-corrected two-sided p < 0.05. (D) Robust regression of coupling strength at C4 and task proficiency of the original sample. (E) Same conventions as in (C) but for overnight learning curve change. (F) Same conventions as in (D) but for overnight learning curve change.

In more detail, increase in task proficiency resulted in flattening of the learning curve when removing outliers (Author response image 1, rho_s_ = -0.70, p < 0.001) and when applying robust regression analysis (Author response image 1, b = -0.30, t(67) = -10.89, rho = -0.80, p < 0.001). Likewise, higher coupling strength still predicted better task proficiency (mean rho = 0.35, p = 0.029, cluster-corrected) and flatter learning curves after sleep (rho = -0.44, p = 0.047, cluster-corrected) when removing the outliers (Author response image 1) and when calculating robust regression (Author response image 1), task proficiency: (b = 82.32, t(40) = 3.12, rho = 0.45, p = 0.003; learning curve: b = -26.84, t(40) = -2.96, rho = -0.43, p = 0.005). Furthermore, we calculated spearman rank correlations and cluster-corrected spearman rank correlations in our original manuscript, to mitigate the impact of outliers, even though Pearson correlations are more widely used in the field. Therefore, we still report spearman rank correlations for single electrodes instead of robust correlations as it is more consistent with the cluster-correlation analyses.

We now use robust trend lines instead of linear trend lines in our scatter plots. Further, we added the correlations without outliers (Author response image 1) to the supplements as Figure 2 —figure supplement 1D and Figure 3 —figure supplement 2 FG. These additional analyses are now reported in the Results section of the revised manuscript (page 9, lines 186 – 191):

“we confirmed a strong negative correlation between the change (post retention values – pre retention values) in task proficiency and the change in learning curve after the retention interval (Figure 2F; rho_s_ = -0.71, p < 0.001), which also remained strong after outlier removal (Figure 2 —figure supplement 1D). This result indicates that participants who consolidate their juggling performance after a retention interval show slower gains in performance.”

And (page 16, lines 343 – 346):

“Furthermore, our results remained consistent when including coupled spindle events in NREM2 (Figure 3 —figure supplement 2E) and after outlier removal (Figure 3 —figure supplement 2FG).”

Furthermore, we now state that we specifically utilized spearman rank correlations to mitigate the impact of outliers in our analyses in the method section (page 35, lines 808 – 813):

“For correlational analyses we utilized spearman rank correlations (rho_s_; Figure 2F and Figure 3DE) to mitigate the impact of possible outliers as well as cluster-corrected spearman rank correlations by transforming the correlation coefficients to t-values (p < 0.05) and clustering in the space domain (Figure 3DE). Linear trend lines were calculated using robust regression.”

5. With only a single night of recording data, it is impossible to disentangle possible trait-based sleep characteristics (e.g., Subject 1 has high SO-spindle coupling in general and retains motor memories well, but these are independent of each other) from a specific, state-based account (e.g., Sub’ect 1's high SO-spindle coupling on night 1 specifically led to their improved retention or change in learning, etc., and this is unrelated to their general SO-spindle coupling or motor performance abilities). Clearly, many studies face this limitation, but this should be acknowledged.

We thank the reviewers for their important remark. We agree that it is impossible to make a sound statement about whether our reported correlations represent trait- or state-based aspects of the sleep and learning relationship with the data that we have reported in the manuscript. However, while we are lacking a proper baseline condition without any task engagement, we still recorded polysomnography for all subjects during an adaptation night. Given the expected pronounced differences in sleep architecture between the adaptation nights and learning nights (see Supplementary file 1 – table 1 for an overview collapsed across both age groups), we initially refrained from entering data from the adaptation nights into our original analyses, but we now fully report the data below. Note that the differences are driven by the adaptation night, where subjects first have to adjust to sleeping with attached EEG electrodes in a sleep laboratory.

To further clarify whether subjects with high coupling strength have a motor learning advantage (i.e. trait-effect) or a learning induced enhancement of coupling strength is indicative for improved overnight memory change (i.e. state-effect), we ran additional analyses using the data from the adaptation night. Note that the coupling strength metric was not impacted by differences in event number and our correlations with behavior were not influenced by sleep architecture (please refer to our answer of issue #7 for the results).Therefore, we considered it appropriate to also utilize data from the adaptation night.

First, we correlated SO-spindle coupling strength obtained from the adaptation night with the coupling strength in the learning night. We found that overall, coupling strength is highly correlated between the two measurements (mean rho across all channels = 0.55), supporting the notion that coupling strength remains rather stable within the individual (i.e. trait), similar to what has been reported about the stable nature of sleep spindles as a “neural finger-print” (De Gennaro and Ferrara, 2003; De Gennaro et al., 2005; Purcell et al., 2017).

To investigate a possible state-effect for coupling strength and motor learning, we calculated the difference in coupling strength between the two nights (learning night – adaptation night) and correlated these values with the overnight change in task proficiency and learning curve. We identified no significant correlations with a learning induced coupling strength change; neither for task proficiency nor learning curve change. Note that there was a positive correlation of coupling strength change with overnight task proficiency change at Cz, however it did not survive cluster-corrected correlational analysis (rho_s_ = 0.34, p = 0.15). Combined, these results favor the conclusion that our correlations between coupling strength and learning rather reflect a trait-like relationship than a state-like relationship. This is in line with the interpretation of our previous studies that SO-spindle coupling strength reflects the efficiency and integrity of the neuronal pathway between neocortex and hippocampus that is paramount for memory networks and the information transfer during sleep (Hahn et al., 2020; Helfrich et al., 2019; Helfrich et al., 2018; Winer et al., 2019). For a comprehensive review please see Helfrich et al. (2021), which argued that SO-spindle coupling predicts the integrity of memory pathways and therefore correlates with various metrics of behavioral performance or structural integrity.

We have now added the additional state-trait analyses to the updated manuscript as Figure 3 —figure supplement 2HI and report them in the Results section (page 17, lines 361 – 375):

“Finally, we investigated whether subjects with high coupling strength have a gross-motor learning advantage (i.e. trait-effect) or a learning induced enhancement of coupling strength is indicative for improved overnight memory change (i.e. state-effect). First, we correlated SO-spindle coupling strength obtained from the adaptation night with the coupling strength in the learning night. We found that overall, coupling strength is highly correlated between the two measurements (mean rho across all channels = 0.55, Figure 3 —figure supplement 2H), supporting the notion that coupling strength remains rather stable within the individual (i.e. trait). Second, we calculated the difference in coupling strength between the learning night and the adaptation night to investigate a possible state-effect. We found no significant cluster-corrected correlations between coupling strength change and task proficiency- as well as learning curve change (Figure 3 —figure supplement 2I).”

Collectively, these results indicate the regionally specific SO-spindle coupling over central EEG sensors encompassing sensorimotor areas precisely indexes learning of a challenging motor task.

We further refer to these new results in the Discussion section (page 23, lines 521 – 528):

“Moreover, we found that SO-spindle coupling strength remains remarkably stable between two nights, which also explains why a learning-induced change in coupling strength did not relate to behavior (Figure 3 —figure supplement 2I). Thus, our results primarily suggest that strength of SO-spindle coupling correlates with the ability to learn (trait), but does not solely convey the recently learned information. This set of findings is in line with recent ideas that strong coupling indexes individuals with highly efficient subcortical-cortical network communication (Helfrich et al., 2021).”

Additionally, we now provide descriptive data of the adaptation and learning night in the Supplementary file – table 1 and explicitly mention the adaptation night in the Results section, which was previously only mentioned in the method section (page 6, lines 101 – 105):

“Polysomnography (PSG) was recorded during an adaptation night and during the respective sleep retention interval (i.e. learning night) except for the adult wake-first group (for sleep architecture descriptive parameters of the adaptation night and learning night as well as for adolescents and adults see Supplementary file – table 1 and 2).”

6. The authors used a partial correlation analysis to rule out that age drove the relationship between coupling strength, learning curve and task proficiency. It seems like this analysis was done specifically for electrode C4, after having already established that coupling strength at electrode C4 correlates in general with changes in the learning curve and task proficiency. The claim that results were not driven by age as confounding factor would be stronger if the authors used a cluster-corrected partial correlation in the first place (just as in the main analysis).

The reviewers are correct that initially we only conducted the partial correlation for electrode C4. Following the reviewers suggestion we now additionally computed cluster-corrected partial correlations similar to our main analysis. Like in our original analyses, we found a significant positive central cluster (mean rho = 0.40, p = 0.017) showing that higher coupling strength related to better task proficiency after sleep and a negative cluster-corrected correlation at C4 showing that higher coupling strength was related to flatter learning curves after sleep (rho = -0.47, p = 0.049) also when controlling for age.

We now always report cluster-corrected partial correlations when controlling for possible confounding variables in the updated version of the manuscript (also see answer to issue #7). A summary of all computed partial correlations can now be found as Figure 3 —figure supplement 3 and 4 in the revised manuscript.

Specifically we now state in the Results section (page 16 – 17, lines 347 – 360):

“To rule out age as a confounding factor that could drive the relationship between coupling strength, learning curve and task proficiency in the mixed sample, we used cluster-corrected partial correlations to confirm their independence of age differences (task proficiency: mean rho = 0.40, p = 0.017; learning curve: rho_s_ = -0.47, p = 0.049). Additionally, given that we found that juggling performance could underlie a circadian modulation we controlled for individual differences in alertness between subjects due to having just slept. We partialed out the mean PVT reaction time before the juggling performance test after sleep from the original analyses and found that our results remained stable (task proficiency: mean rho = 0.37, p = 0.025; learning curve: rho_s_ = -0.49, p = 0.040). For a summary of the reported cluster-corrected partial correlations as well as analyses controlling for differences in sleep architecture see Figure 3 —figure supplement 3. Further, we also confirmed that our correlations are not influenced by individual differences in SO and spindle event parameters (Figure 3 —figure supplement 4).”

And in the methods section (page 35, lines 813 – 814):

“To control for possible confounding factors we computed cluster-corrected partial rank correlations (Figure 3 —figure supplement 3 and 4).”

7. To allow a more comprehensive assessment of the underlying data information with regards to general sleep descriptives (minutes, per cent of time spent in different sleep stages, overall sleep time etc.) as well as related to SOs, spindles and coupled events (e.g. number, density etc.) would be needed.

We agree with the reviewers that additional information about sleep architecture and SO as well as sleep spindle characteristics are needed for a more comprehensive assessment of our data. We now added summary tables for sleep architecture and SO/spindle event descriptive measures for the whole sample and for the sleep first groups that we used for our correlational analyses to the supplementary material in the updated manuscript. It is important to note, that due to the longer sleep opportunity of adolescents that we provided to accommodate the overall higher sleep need in younger participants, adolescents and adults differed in most general sleep architecture markers and SO as well as sleep spindle descriptive measures. In addition, changes in sleep architecture are prominent during the maturational phase from adolescence to adulthood, which might introduce additional variance between the two age groups.

We now provide general sleep descriptives in the revised version of the manuscript as Supplementary file – table 2 and table 7. These data are referred to in the Results section (page 6, lines 101 – 105):

“Polysomnography (PSG) was recorded during an adaptation night and during the respective sleep retention interval (i.e. learning night) except for the adult wake-first group (for sleep architecture descriptive parameters of the adaptation night and learning night as well as for adolescents and adults see Supplementary file – table 1 and 2).”

And (page 15, lines 311 – 318):

“Furthermore, given that we only recorded polysomnography for the adults in the sleep first group and that adolescents in the wake first group showed enhanced task proficiency at the time point of the sleep retention interval due to additional training (Figure 3 —figure supplement 2A), we only considered adolescents and adults of the sleep-first group to ensure a similar level of juggling experience (for summary statistics of sleep architecture and SO and spindle events of subjects that entered the correlational analyses see Supplementary file – table 7).”

The additional control analyses are also now added to the revised manuscript as Figure 3 —figure supplement 3 and 4 in the Results section (page 16, lines 356 – 360):

“For a summary of the reported cluster-corrected partial correlations as well as analyses controlling for differences in sleep architecture see Figure 3 —figure supplement 3. Further, we also confirmed that our correlations are not influenced by individual differences in SO and spindle event parameters (Figure 3 —figure supplement 4).”

8. The authors state “that "To ensure the simultaneous presence of the two interacting sleep oscillations in the signal, we restricted our analyses to NREM3 sleep given the higher co-occurrence” rate." We do not understand this reasoning. The utilized procedure of specifically isolating sleep spindles that are followed or preceded by slow oscillations already ensures the presence of SOs and sleep spindles in the data. Hence, why not take coupled events from sleep stage N2 into account? Or do the authors think that light sleep SO-spindle events are qualitatively different from SWS SO-spindle complexes (and if so does the present data support such a notion)?

We thank the reviewers for bringing this issue to our attention and apologize for not explaining our rationale about only including coupled NREM3 events in our analyses clearly enough. First, to our knowledge, there is no consensus yet about whether SO-spindle complexes differ between light and deep sleep. Instead evidence rather points to a distinction between coupled and uncoupled spindle events due to their differences in underlying cortical circuitry dynamics (Niethard et al., 2018) and hippocampal-neocortical communication (Helfrich et al., 2019).

Accordingly, the reviewers are correct, that given that we control for co-occurrence to ensure concomitant SO and spindle oscillations, a perfectly viable approach would be to analyze all coupled events instead of just events in NREM3. However, we decided against this approach in our initial analyses because of two reasons. First, similar to our previous longitudinal study (Hahn et al., 2020), we found that overall SO-spindle co-occurrence (%) is extremely low in NREM2 sleep compared to NREM3 sleep (Figure R9A, sleep stage main effect: F(1, 51) = 1209.09, p < 0.001, partial eta² = 0.96, age main effect: F(1, 51) = 11.35, p = 0.001, partial eta² = 0.18, interaction: F(1, 51) = 0.02, p = 0.89, partial eta² < 0.001). Further, some subjects did not even show co-occurring SO-spindle events in a handful of electrodes during NREM2 (Figure R9B, data collapsed across all subjects and electrodes). Importantly, we showed earlier that the low co-occurrence rate, or in other words, the high amount of isolated spindles in NREM2 can introduce a serious amount of noise for the preferred SO-phase estimation (Figure R9C) which eventually can mask a potential behavioral correlate (Figure R9D). Therefore, we opted for a conservative approach for our analyses by only computing coupling strength for co-occurring events in NREM3. The second reason why we did not include coupled events during NREM2 sleep is that we wanted to apply the exact same analysis pipeline we developed in the previous study to the current research advance study. We are convinced that demonstrating the behavioral relevance of SO-spindle coupling in a different age sample performing an ecologically valid motor task by employing the same signal processing pipeline is one key strength of the present research advance manuscript.

Regardless, we recalculated SO-spindle coupling strength for co-occurrence controlled events across both sleep stages and correlated the results with task proficiency change and learning curve change after sleep. Similar to our original results higher coupling strength indicated better task proficiency (Figure R9E, left, rho = 0.45, p = 0.047, cluster-corrected) and flatter learning curves after sleep (Figure R9E, right, rho = -0.46, p = 0.037, cluster-corrected). However, given the low co-occurrence rate in NREM2 (cf. Figure R9AB) a reliable estimation of coupling strength is compromised and therefore we cannot give a sound assessment of whether coupled in events in NREM2 differ from coupled events in NREM3.

We added Figure R9AB to the revised manuscript as Figure 3 —figure supplement 1CD and further added a clarifying statement about our rationale of only analyzing events in NREM3 to the Results section (page 12 – 13, lines 258 – 264):

“To ensure the simultaneous presence of the two interacting sleep oscillations in the signal, we followed a conservative approach and restricted our analyses to NREM3 sleep given the low co-occurrence rate in NREM2 sleep (Figure 3 —figure supplement 1CD) which can cause spurious coupling estimates (Hahn et al., 2020). Further, we only considered spindle events that displayed a concomitant detected SO within a 2.5 s time window.”

And in the methods section (page 34, lines 763 – 770):

“Ensuring co-occurrence of SO and sleep spindles

Cross-frequency coupling renders meaningful information of network communication only when the suspected interacting oscillations are present in the signal. Therefore, we only analyzed SO and sleep spindle epochs during which they co-occurred in a 2.5s time window (± ~2 SO cycles around the spindle peak). Furthermore, we restricted all our coupling analyses to sleep stage NREM3 because of general lower co-occurrence of SO and spindles in NREM2 (Figure 3 —figure supplement 1CD), which can cause spurious coupling estimates (Hahn et al., 2020).”

The additional correlational analyses that include co-occurrence corrected events in NREM2 (Figure R9E) can now be found in the Results section as Figure 3 —figure supplement 2FG in the Results section (page 16, lines 343 – 346):

“Furthermore, our results remained consistent when including coupled spindle events in NREM2 (Figure 3 —figure supplement 2E) and after outlier removal (Figure 3 —figure supplement 2FG).”

[Editors' note: further revisions were suggested prior to acceptance, as described below.]

The manuscript has been improved but there are some remaining issues that need to be addressed, as outlined below:The authors have done an impressive job with this revision. It is meticulously organized, thorough, and clearly stated. That is all to their major credit. However, I still cannot come to agree that their data supports much of the story they are telling.

We thank the reviewers for their positive assessment of our previous work. We appreciate the thoughtful and detailed feedback on how to improve the main message of our manuscript.

First, part of the issue may be the change from their original story and the new one following the revision. Making major revisions can obviously be tricky, especially when a revision requires as many changes as theirs did (and I again commend them on the overhaul). But there is still something unclear in their primary claims. They say in their cover letter, "Collectively, our results suggest that SO-spindle coupling indexes the integrity of memory pathways (as reviewed in detail recently: Helfrich et al., 2021); thus, reflects a trait-specific (in contrast to a state-specific) correlate of learning capacity." However, this story does not come through clearly in the new paper. In fact, reading the new paper, it seems this is nodded to only here in the Discussion“ – "Thus, our results primarily suggest that strength of SO-spindle coupling correlates with the ability to learn (trait), but does not solely convey 534 the recently learned information. This set of findings is in line with recent ideas that strong coupling indexes individuals with highly efficient subcortical-cortical network communication (Helfrich et al., ”021)." Much of the paper instead talks about active systems consolidation theory, which I believe is not supported in their data, and the authors do seem to agree. If the authors indeed want to make this more of a memory pathway integrity story, it seems more unpacking of the ideas in their recent review is warranted, as does perhaps some evidence in the literature linking sleep measures to integrity in some neural pathways (e.g., Mander et al., 2017).

We thank the reviewer for making us aware of this issue. We agree that the neural efficiency literature needs to be more prominently featured in our manuscript. Thus, we incorporated references and extended discussion in multiple instances in the manuscript. To highlight this consideration from the very beginning, we added a new paragraph to the introduction in the revised version of the manuscript (pages 3 -4, lines 40 – 50):

“Several lines of research recently demonstrated that precisely timed SO-spindle interaction mediates successful memory consolidation across the lifespan (Hahn et al., 2020; Helfrich et al., 2018; Mikutta et al., 2019; Molle et al., 2011; Muehlroth et al., 2019). Critically, SO-spindle coupling as well as spindles and SOs in isolation are related to neural integrity of memory structures such as medial prefrontal cortex, thalamus, hippocampus and entorhinal cortex (Helfrich et al., 2021; Helfrich et al., 2018; Ladenbauer et al., 2017; Mander et al., 2017; Muehlroth et al., 2019; Spano et al., 2020; Winer et al., 2019). Thus converging evidence suggests that SO-spindle coupling does not only actively transfer mnemonic information during sleep but also indexes general efficiency of memory pathways (Helfrich et al., 2021; Mander et al., 2017).”

Furthermore, we now critically discuss the active contribution of sleep to gross-motor memory learning in the discussion (pages 23 – 24, lines 522 – 536):

“How ‘active’ is sleep for real-life gross-motor memory consolidation? We found that sleep impacts the learning curve but did not affect task proficiency in comparison to a wake retention interval directly after learning (Figure 2DE). Three accounts might explain the absence of a sleep effect on task proficiency. (1) Sleep rather stabilizes than improves gross-motor memory, which is in line with previous gross-motor adaption studies (Bothe et al., 2019; Bothe et al., 2020). This parallels findings in finger tapping tasks were the narrative evolved from sleep-related performance improvements (Walker et al., 2002) to stabilization (Brawn et al., 2010). (2) Pre-sleep performance is critical for sleep to improve motor skills (Wilhelm et al., 2012). Participants commonly reach asymptotic pre-sleep performance levels in finger tapping tasks, which is most frequently used to probe sleep effects on motor memory. Here we found that using a complex juggling tasks, participants do not reach asymptotic ceiling performance levels in such a short time. Indeed, the learning progression for the sleep-first and wake-first groups followed a similar trend (Figure 2AB), suggesting that more training and not in particular sleep drove performance gains.”

Additionally, we would like to emphasize that state and trait effects are not mutually exclusive. In fact, the overlaps of state and trait effects have been previously identified in sleep spindle literature (Lustenberger et al., 2015; Schabus et al., 2006). With SO-spindle coupling we do indeed investigate the mechanistic assumption of the active systems memory consolidation theory. However, like spindles, SO-spindle coupling relates to memory reactivation (cf. Cairney et al., 2018; Schreiner et al., 2021) and also to neural integrity/efficiency (Helfrich et al., 2018; Muehlroth et al., 2019). Thus, memory consolidation does not only need an information transfer between memory structures in the brain (i.e. Cortex and Hippocampus) but it also needs an efficient neural pathway to exchange said information.

We now also elaborate on this issue in the discussion (page 25, lines 550 – 561):

“Importantly, SO-spindle coupling still predicted learning dynamics on a single subject level advocating for a supportive function of sleep for gross-motor memory. Moreover, we found that SO-spindle coupling strength remains remarkably stable between two nights, which also explains why a learning-induced change in coupling strength did not relate to behavior (Figure 3 —figure supplement 2I). Thus, our results primarily suggest that strength of SO-spindle coupling correlates with the ability to learn (trait), but does not solely convey the recently learned information. Note that state and traits effects are not mutually exclusive. The overlap of state and trait effects is a long-standing issue in spindle literature, which also seems so apply to their coordinated interplay with SOs (Lustenberger et al., 2015; Schabus et al., 2006). This set of findings is in line with recent ideas that strong coupling indexes individuals with highly efficient subcortical-cortical network communication (Helfrich et al., 2021).”

Second, they concede in various locations that the circadian story cannot be ruled out, which I also commend, but then the paper still largely revolves around active sleep consolidation theory. I invite the authors to imagine convincing a hypothetical researcher who thought the brain just shuts off entirely during sleep ("sleep does nothing") and that people have different abilities based on the time of day. (Believe it or not, this is not my belief.) How would the authors convince this person based on these behavioral data that sleep is actually doing something? I do not know whether they could, given the mixed-effects model findings.Of course, they could point to the prior literature. The prior literature on sleep and motor learning has shown, in the case of the Morita juggling studies cited, that there should be better overall performance after sleep (vs equivalent wake periods). And in the case of countless finger-tapping studies, even though the major story has changed from one of absolute improvement (e.g., Walker et al., 2002) to stabilization (e.g., Brawn et al., 2010) after sleep, there seem to be sleep (vs wake) benefits on overall performance (analogous to task proficiency here). This, however, is not what the authors find with their learning curve findings here, as performance seems, if anything, worse on the first few trials after sleep (though this may not be significant) and then catches up more quickly. So, it is hard to know whether the prior literature would necessarily help them convince this researcher about their own findings.This researcher may also say that the inclusion of a PVT is great, and the null results across sessions is more helpful than not to their story. But this researcher may add that a null PVT difference does not exclude all possible circadian effects. There are certainly circadian effects on cognition – including the very recent publication of Tandoc et al., (2021) and even on motor learning (Keisler et al., 2007) – and indeed the authors do find such an effect here in their mixed-effects model analysis. Therefore, the null PVT results are not conclusive, especially in counteracting an effect that they actually found in their paper.One could then point this researcher to the SO-spindle coupling results as evidence that sleep is playing a strong role here. However, given that these are trait- vs. state-based results, it is unclear why stronger SO-spindle coupling for some individuals – which may be having an impact on neural integrity over a long timescale – would prime their nervous systems for more learning right after sleep than at some other time during the day. The researcher may say, okay, SO-spindle coupling results do not prove sleep does anything, they merely correlate with the observed behavioral result, and moreover, they constitute a trait (vs post-learning sleep state) effect. They may add that it is also unclear why, if stronger SO-spindle coupling is doing something, it could not alternatively reflect some other individual trait that could even lead to the observed circadian effects that learning curves are higher in the morning.One analysis that could possibly work to disentangle circadian vs. active sleep effects would be to include a different factor in the mixed-effects model that could tease apart time of day from sleep-after-learning effects. In addition to including "Time of day", where all mornings = 1 and all evenings = 0, the authors could include the conjunction of "Time of day + after learning", where mornings on the 2nd and 3rd sessions = 1 and mornings on the 1st session and all evenings = 0. This would capture the idea that post-learning mornings show differential improvement because post-learning sleep sort of "prepared" the networks to re-learn within a short time span, and this preparation was not operative before the 1st session. I say it could "possibly" work above because the two factors would still be quite correlated (identical except for the first morning session), which could hurt their statistical power to independently produce effects. Nevertheless, if BOTH factors end up being significant, I think the authors could make the claim that both are contributing (that is, time of day + after learning is actually independently contributing above and beyond what time of day could do alone). If only one is significant, then the story is clean, but may have to change. If neither are significant, then it may be difficult to know what to do, and the authors may have to fall back on the original time-of-day analysis and keep things closer to as is but acknowledge more of the uncertainty surrounding the effects. If nothing changes upon a second revision in this regard, I do expect the authors to incorporate circadian possibilities more thoroughly in their paper, such as in their abstract and with more citations of this literature.I realize this may seem a lot for a second round of revisions, and the authors have clearly done an impressive amount of work on the paper, but I feel that the authors can still strengthen it, either with this last analysis or by refocusing on the stories that can and cannot be supported here. There is something here that lacks clarity in translating from the data to the story about them, and, as a result, it remains difficult to confidently find the main takeaway from the manuscript.

We thank the reviewer for the detailed feedback and the valuable input on subsequent analyses and interpretations. We followed the reviewers’ suggestion and calculated additional mixed-models to further disentangle sleep and time of day effect. We modeled learning curve and task proficiency separately across all testing blocks with the fixed effects “time of day” and “sleep after learning” (i.e. performance test 2 for the sleep first group and performance test 3 for the wake first group) and subjects as random effects (see Table 6 for the full report).

For learning curve, we found that the fixed effect “time of day” approached conventional significance levels (time of day: Β = -1.008, t(202) = -1.625, p = 0.106, CI95 = [-2.231, 0.215]) using the suggested mixed-effect model. Sleep after learning had no effect on the learning curve (Β = 0.172, t(202) = 0.268, p = 0.789, CI95 = [-1.093, 1.437]). Task proficiency however, was overall better in the evening performance tests (Β = 5.751, t(202) = 2.252, p = 0.011, CI95 = [1.310, 10.192]) and additionally benefited from sleep after learning (Β = 3.795, t(202) = 1.672, p = 0.096, CI95 = [-0.680, 8.271]).

The reviewer correctly noted that the correlative nature of the fixed effects might dilute statistical power to prevent the identification of independent effects. However, given the lower end of the confidence interval close to zero for the fixed effect “sleep after learning” there is not enough evidence to accept the null hypotheses that sleep has no effect at all. Consequently, it suggests that both time of day and sleep seem to contribute to the overall juggling performance.

We have now added table 6 to the supplementary file 1 and updated the Results section with these important additional analyses (pages 9 – 10, lines 189 – 200):

“However, these analyses cannot exclude all circadian effects. Therefore, we modeled learning curve and task proficiency with time of day (morning session, evening session) and sleep after learning as fixed effects and subjects as random effects to further disentangle circadian and sleep specific effects. Results for learning curve were inconclusive for both fixed effects (time of day: Β = -1.008, t(202) = -1.625, p = 0.106, CI95 = [-2.231, 0.215]; Sleep after learning: Β = 0.172, t(202) = 0.268, p = 0.789, CI95 = [-1.093, 1.437]; Supplementary file 1 – table 6A). Task proficiency was overall better in the evening performance tests (Β = 5.751, t(202) = 2.252, p = 0.011, CI95 = [1.310, 10.192]) and additionally trended to benefit from sleep after learning (Β = 3.795, t(202) = 1.672, p = 0.096, CI95 = [-0.680, 8.271]; Supplementary file 1 – table 6B). These results suggest that both, time of day and sleep contribute to the overall juggling performance.”

In the light of these additional analyses and feedback from the reviewer, we incorporated possible time of day effects more thoroughly in the updated version of the manuscript.

We now report time of day effects in the abstract (page 2, lines 1 – 16):

“Previously, we demonstrated that precise temporal coordination between slow oscillations (SO) and sleep spindles indexes declarative memory network development (Hahn et al., 2020). However, it is unclear whether these findings in the declarative memory domain also apply in the motor memory domain. Here, we compared adolescents and adults learning juggling, a real-life gross-motor task. Juggling performance was impacted by sleep and time of day effects. Critically, we found that improved task proficiency after sleep lead to an attenuation of the learning curve, suggesting a dynamic juggling learning process. We employed individualized cross-frequency coupling analyses to reduce inter and intra-group variability of oscillatory features. Advancing our previous findings, we identified a more precise SO-spindle coupling in adults compared to adolescents. Importantly, coupling precision over motor areas predicted overnight changes in task proficiency and learning curve, indicating that SO-spindle coupling is relates to the dynamic motor learning process. Our results provide first evidence that regionally specific, precisely coupled sleep oscillations support gross-motor learning.”

The new results are also further elaborated in the Discussion section. Additionally, we now fully acknowledge that the null-effect of the PVT does not exclude all possible circadian effects (pages 24 – 25, lines 536 – 549):

“How ‘active’ is sleep for real-life gross-motor memory consolidation? […] (3) Sleep effects are intermingled with time of day effects on juggling performance. Indeed, the steeper learning curve after the first retention interval in the sleep first group can also be interpreted as a time of day effect. However, when modeling time of day and sleep specific effects across all performance blocks, we found a trend that sleep after learning supports task proficiency. Note, that the correlative nature of both factors in the model likely resulted in insufficient statistical power to produce independently significant results. Additionally, we did not find evidence for a circadian modulation of cognitive engagement based on objective reaction time data in our study (Figure 2 —figure supplement 1C). However, a null-result does not exclude all possible circadian effects and ample evidence suggests that cognitive performance and motor learning are influenced by the time of day (Blatter and Cajochen, 2007; Keisler et al., 2007; Tandoc et al., 2021). Specifically, implicit learning seems to be affected by time of day rather than sleep (Keisler et al., 2007). Therefore, we cannot fully disentangle circadian and sleep effects with our study design, which should be considered a limitation to our findings.”

We now also reference the time of day effects in the conclusion (page 26, lines 577 – 583):

“Taken together, our results provide a mechanistic understanding of how the brain forms real-life gross-motor memory during sleep. However, how time of day additionally affects and interacts with sleep to support gross-motor learning remains an open question. As sleep has been shown to support fine-motor memory consolidation in individuals after stroke (Gudberg and Johansen-Berg, 2015; Siengsuhon and Boyd, 2008), SO-spindle coupling integrity could be a valuable, easy to assess predictive index for rehabilitation success.”